# TEST TIME LEARNING FOR TIME SERIES FORECASTING

## ABSTRACT

Time-series forecasting has seen significant advancements with the introduction of token prediction mechanisms such as multi-head attention. However, these methods often struggle to achieve the same performance as in language modeling, primarily due to the quadratic computational cost and the complexity of capturing long-range dependencies in time-series data. State-space models (SSMs), such as Mamba, have shown promise in addressing these challenges by offering efficient solutions with linear RNNs capable of modeling long sequences with larger context windows. However, there remains room for improvement in accuracy and scalability.

We propose the use of Test-Time Training (TTT) modules in a parallel architecture to enhance performance in long-term time series forecasting. Through extensive experiments on standard benchmark datasets, we demonstrate that TTT modules consistently outperform state-of-the-art models, including the Mamba-based TimeMachine, particularly in scenarios involving extended sequence and prediction lengths. Our results show significant improvements in Mean Squared Error (MSE) and Mean Absolute Error (MAE), especially on larger datasets such as Electricity, Traffic, and Weather, underscoring the effectiveness of TTT in capturing long-range dependencies. Additionally, we explore various convolutional architectures within the TTT framework, showing that even simple configurations like 1D convolution with small filters can achieve competitive results. This work sets a new benchmark for time-series forecasting and lays the groundwork for future research in scalable, high-performance forecasting models.

## 1 INTRODUCTION

Long Time Series Forecasting (LTSF) is a crucial task in various fields, including energy Doe & Smith (2023), industry Doe & Smith (2024), defense Bakdash et al. (2017), and atmospheric sciences Lim & Zohren (2020). LTSF uses a historical sequence of observations, known as the look-back window, to predict future values through a learned or mathematically induced model. However, the stochastic nature of real-world events makes LTSF challenging. Deep learning models, including time series forecasting, have been widely adopted in engineering and scientific fields. Early approaches employed Recurrent Neural Networks (RNNs) to capture long-range dependencies in sequential data like time series. However, recurrent architectures like RNNs have limited memory retention, are difficult to parallelize, and have constrained expressive capacity. Transformers (Vaswani et al., 2017), with ability to efficiently process sequential data in parallel and capture contextual information, have significantly improved performance on time series prediction task (Wen et al., 2023; Liu et al., 2022b; Ni et al., 2024; Chen et al., 2024). Yet, due to the quadratic complexity of attention mechanisms with respect to the context window (or look-back window in LTSF), Transformers are limited in their ability to capture very long dependencies.

In recent years, State Space Models (SSMs) such as Mamba (Gu & Dao, 2024), a gated linear RNN variant, have revitalized the use of RNNs for LTSF. These models efficiently capture much longer dependencies while reducing computational costs and enhancing expressive power and memory retention. A new class of Linear RNNs, known as Test Time Training (TTT) modules (Sun et al., 2024), has emerged. These modules use expressive hidden states and provide theoretical guarantees for capturing long-range dependencies, positioning them as one of the most promising architectures for LTSF and due to their weight adaptation during test time are very effective on non-stationary data. We provide more motivation on TTT for non-stationary data in Appendix A.

## KEY INSIGHTS AND RESULTS

Through our experiments, several key insights emerged regarding the performance of TTT modules when compared to existing SOTA models:

- **Superior Performance with Longer Sequence and Prediction Lengths:** TTT modules consistently outperformed the SOTA TimeMachine model, particularly as sequence and prediction lengths increased. Architectures such as Conv Stack 5 demonstrated their ability to capture long-range dependencies more effectively than Mamba-based models, resulting in noticeable improvements in Mean Squared Error (MSE) and Mean Absolute Error (MAE) across various benchmark datasets.

- **Strong Improvement on Larger Datasets:** On larger datasets, such as Electricity, Traffic, and Weather, the TTT-based models excelled, showing superior performance compared to both Transformer- and Mamba-based models. These results underscore the ability of TTT to handle larger temporal windows and more complex data, making it especially effective in high-dimensional, multivariate datasets.

- **Hidden Layer Architectures:** The ablation studies revealed that while convolutional architectures added to the TTT modules provided some improvements, Conv Stack 5 consistently delivered the best results among the convolutional variants. However, simpler architectures like Conv 3 often performed comparably, showing that increased architectural complexity did not always lead to significantly better performance. Very complex architectures like the modern convolutional block from Donghao & Xue (2024) showed competitive performance when used as TTT hidden layer architectures compared to the simpler single architectures proposed, hinting on the potential of more complex architectures in capturing more long term dependencies.

- **Adaptability to Long-Term Predictions:** The TTT-based models excelled in long-term prediction tasks, especially for really high prediction lengths like 2880. TTT-based models also excelled on increased sequence lengths as high as 5760 which is the maximum sequence length allowed by the benchmark datasets. This verified the theoretically expected superiority of TTT based models relative to the mamba/transformer based SOTA models.

## MOTIVATION AND CONTRIBUTIONS

In this work, we explore the potential of TTT modules in Long-Term Series Forecasting (LTSF) by integrating them into novel model configurations to surpass the current state-of-the-art (SOTA) models. Our key contributions are as follows:

- We propose a new model architecture utilizing quadruple TTT modules, inspired by the TimeMachine model (Ahamed & Cheng, 2024), which currently holds SOTA performance. By replacing the Mamba modules with TTT modules, our model effectively captures longer dependencies and predicts larger sequences.

- We evaluate the model on benchmark datasets, exploring the original look-back window and prediction lengths to identify the limitations of the SOTA architecture. We demonstrate that the SOTA model achieves its performance primarily by constraining look-back windows and prediction lengths, thereby not fully leveraging the potential of LTSF.

- We extend our evaluations to significantly larger sequence and prediction lengths, showing that our TTT-based model consistently outperforms the SOTA model using Mamba modules, particularly in scenarios involving extended look-back windows and long-range predictions.

- We conduct an ablation study to assess the performance of various hidden layer architectures within our model. By testing six different convolutional configurations, one of which being ModernTCN by Donghao & Xue (2024), we quantify their impact on model performance and provide insights into how they compare with the SOTA model.

## 2 RELATED WORK

**Transformers for LTSF** Several Transformer-based models have advanced long-term time series forecasting (LTSF), with notable examples like iTransformer (Liu et al., 2024) and PatchTST

(Nie et al., 2023). iTransformer employs multimodal interactive attention to capture both temporal and inter-modal dependencies, suitable for multivariate time series, though it incurs high computational costs when multimodal data interactions are minimal. PatchTST, inspired by Vision Transformers (Dosovitskiy et al., 2021), splits input sequences into patches to capture dependencies effectively, but its performance hinges on selecting the appropriate patch size and may reduce model interpretability. Other influential models include Informer (Zhou et al., 2021), which uses sparse self-attention to reduce complexity but may overlook finer details in multivariate data; Autoformer (Wu et al., 2022), which excels in periodic data but struggles with non-periodic patterns; Pyraformer (Liu et al., 2022b), which captures multi-scale dependencies through a hierarchical structure but at the cost of increased computational requirements; and Fedformer (Zhou et al., 2022), which combines time- and frequency-domain representations for efficiency but may underperform on noisy time series. While each model advances LTSF in unique ways, they also introduce specific trade-offs and limitations.

**State Space Models for LTSF**   S4 models (Gu et al., 2022a;b; Gupta et al., 2023) are efficient sequence models for long-term time series forecasting (LTSF), leveraging linear complexity through four key components: $\Delta$ (discretization step size), $A$ (state update matrix), $B$ (input matrix), and $C$ (output matrix). They operate in linear recurrence for autoregressive inference and global convolution for parallel training, efficiently transforming recurrences into convolutions. However, S4 struggles with time-invariance issues, limiting selective memory. Mamba (Gu & Dao, 2024) addresses this by making $B$, $C$, and $\Delta$ dynamic, creating adaptable parameters that improve noise filtering and maintain Transformer-level performance with linear complexity. SIMBA (Patro & Agneeswaran, 2024) enhances S4 by integrating block-sparse attention, blending state space and attention to efficiently capture long-range dependencies while reducing computational overhead, ideal for large-scale, noisy data. TimeMachine (Ahamed & Cheng, 2024) builds on these advances by employing a quadruple Mamba setup, managing both channel mixing and independence while avoiding Transformers' quadratic complexity through multi-scale context generation, thereby maintaining high performance in long-term forecasting tasks.

**Linear RNNs for LTSF**   RWKV-TS (Hou & Yu, 2024) is a novel linear RNN architecture designed for time series tasks, achieving O(L) time and memory complexity with improved long-range information capture, making it more efficient and scalable compared to traditional RNNs like LSTM and GRU. Orvieto et al. (2023) introduced the Linear Recurrent Unit (LRU), an RNN block matching the performance of S4 models on long-range reasoning tasks while maintaining computational efficiency. TTT (Sun et al., 2024) layers take a novel approach by treating the hidden state as a trainable model, learning during both training and test time with dynamically updated weights. This allows TTT to capture long-term relationships more effectively through real-time updates, providing an efficient, parallelizable alternative to self-attention with linear complexity. TTT's adaptability and efficiency make it a strong candidate for processing longer contexts, addressing the scalability challenges of RNNs and outperforming Transformer-based architectures in this regard.

**MLPs and CNNs for LTSF**   Recent advancements in long-term time series forecasting (LTSF) have introduced efficient architectures that avoid the complexity of attention mechanisms and recurrence. TSMixer (Chen et al., 2023b), an MLP-based model, achieves competitive performance by separating temporal and feature interactions through time- and channel-mixing, enabling linear scaling with sequence length. However, MLPs may struggle with long-range dependencies and require careful hyperparameter tuning, especially for smaller datasets. Convolutional neural networks (CNNs) have also proven effective for LTSF, particularly in capturing local temporal patterns. ModernTCN (Donghao & Xue, 2024) improves temporal convolution networks (TCNs) using dilated convolutions and a hierarchical structure to efficiently capture both short- and long-range dependencies, making it well-suited for multi-scale time series data.

Building on these developments, we improve the original TimeMachine model by replacing its Mamba blocks with Test-Time Training (TTT) blocks to enhance long-context prediction capabilities. We also explore CNN configurations, such as convolutional stacks, to enrich local temporal feature extraction. This hybrid approach combines the efficiency of MLPs, the local pattern recognition of CNNs, and the global context modeling of TTT, leading to a more robust architecture for LTSF tasks that balances both short- and long-term forecasting needs.

## 3 MODEL ARCHITECTURE

The task of Time Series Forecasting can be defined as follows: Given a multivariate time series dataset with a window of past observations (look-back window) $L$: $(\mathbf{x}_1, \ldots, \mathbf{x}_L)$, where each $\mathbf{x}t$ is a vector of dimension $M$ (the number of channels at time $t$), the goal is to predict the next $T$ future values $(\mathbf{x}_{L+1}, \ldots, \mathbf{x}_{L+T})$.

The TimeMachine (Ahamed & Cheng, 2024) architecture, which we used as the backbone, is designed to capture long-term dependencies in multivariate time series data, offering linear scalability and a small memory footprint. It integrates four Mamba (Gu & Dao, 2024) modules as sequence modeling blocks to selectively memorize or forget historical data, and employs two levels of downsampling to generate contextual cues at both high and low resolutions.

However, Mamba's approach still relies on fixed-size hidden states to compress historical information over time, often leading to the model forgetting earlier information in long sequences. TTT (Sun et al., 2024) uses dynamically updated weights (in the form of matrices inside linear or MLP layers) to compress and store historical data. This dynamic adjustment during test time allows TTT to better capture long-term relationships by continuously incorporating new information. Its Hidden State Updating Rule is defined as:

$$W_t = W_{t-1} - \eta \nabla \ell(W_{t-1}; x_t) = W_{t-1} - \eta \nabla \|f(\tilde{x}_t; W) - x_t\|^2$$

We incorporated TTT into the TimeMachine model, replacing the original Mamba block. We evaluated our approach with various setups, including different backbones and TTT layer configurations. Additionally, we introduced convolutional layers before the sequence modeling block and conducted experiments with different context lengths and prediction lengths. We provide mathematical foundations as to why TTT is able to perform test-time adaptation without catastrophic forgetting and how the module adapts to distribution shifts in Appendix A. In the same Appendix we quantify the computational overhead introduced by test-time updates and provide empirical validation, published by the authors who proposed TTT in Sun et al. (2020), on how it performs on real corrupted data and provide some intuition on the parameter initialization in TTT as discussed in the same reference.

Our goal is to improve upon the performance of the state-of-the-art (SOTA) models in LTSF using the latest advancements in sequential modeling. Specifically, we integrate Test-Time Training (TTT) modules into our model for two key reasons, TTT is theoretically proven to have an extremely long context window, being a form of linear RNN (Orvieto et al., 2023), capable of capturing long-range dependencies efficiently. Secondly, the expressive hidden states of TTT allow the model to capture a diverse set of features without being constrained by the architecture, including the depth of the hidden layers, their size, or the types of blocks used.

### 3.1 GENERAL ARCHITECTURE

Our model architecture builds upon the TimeMachine model (Ahamed & Cheng, 2024), introducing key modifications, as shown in Figure 1a, 1b and 1c. Specifically, we replace the Mamba modules in TimeMachine with TTT (Test-Time Training) modules (Sun et al., 2024), which retain compatibility since both are linear RNNs (Orvieto et al., 2023). However, TTT offers superior long-range dependency modeling due to its adaptive nature and theoretically infinite context window. A detailed visualization of the TTT block and the different proposed architectures can be found in Appendix D

Our model features a two-level hierarchical architecture that captures both high-resolution and low-resolution context, as illustrated in Figure 1a. To adapts to the specific characteristics of the dataset, the architecture handles two scenarios—Channel Mixing and Channel Independence—illustrated in Figure 1b and 1c respectively. A more detailed and mathematical description of the normalization and prediction procedures can be found in Appendix D. We provide a computational complexity analysis of the TTT, Transformer, Mamba and ModernTCN modules in Appendix F and we also provide a computational complexity analysis for our model, TimeMachine, iTransformer, PatchTST, TSMixer and ModernTCN in Appendix G. We also included a mathematical comparison between the Mamba and TTT modules in Appendix B as well as theoretical comparison between the TTT module and models handling noise or temporal regularization in Appendix C.

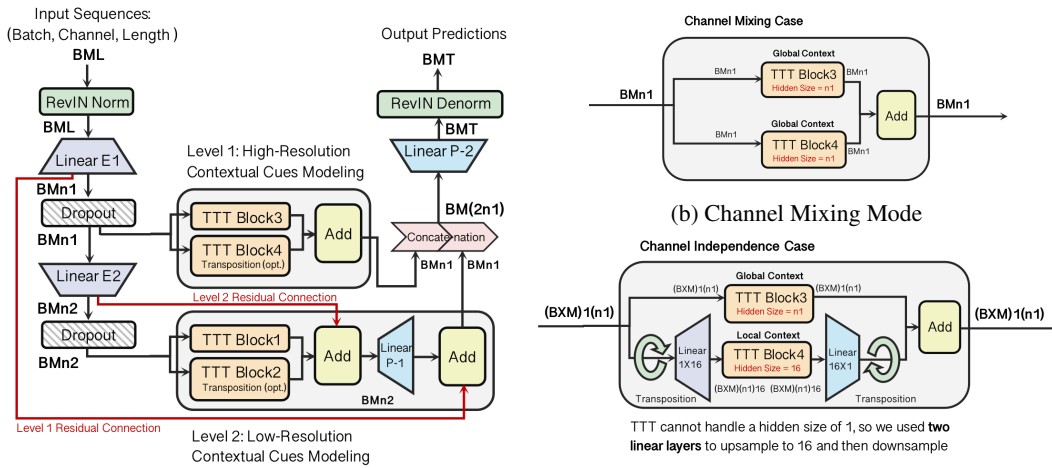

(a) TimeMachine incoporating TTT-Blocks

(b) Channel Mixing Mode

(c) Channel Independence Mode

Figure 1: Our model architecture. (a) We replace the four Mamba Block in TimeMachine with four TTT(Test-Time Training) Block. (b) There are two modes of TimeMachine, the channel mixing mode for capturing strong between-channel correlations, and the channel independence mode for modeling within-channel dynamics. Recent works such as PatchTST (Nie et al., 2023) and TiDE (Das et al., 2024) have shown channel independence can achieve SOTA accuracy. For the channel independence scenario, the inputs are first transposed, and then we integrate two linear layers ($1 \times 16$ and $16 \times 1$) to provide the TTT Block with a sufficiently large hidden size.

## 3.2 HIERARCHICAL EMBEDDING

The input sequence $BML$ (Batch, Channel, Length) is first passed through Reversible Instance Normalization (Kim et al., 2021) (RevIN), which stabilizes the model by normalizing the input data and helps mitigate distribution shifts. This operation is essential for improving generalization across datasets.

After normalization, the sequence passes through two linear embedding layers. Linear E1 and Linear E2 are used to map the input sequence into two embedding levels: higher resolution and lower resolution. The embedding operations $E_1 : \mathbb{R}^{M \times L} \to \mathbb{R}^{M \times n_1}$ and $E_2 : \mathbb{R}^{M \times n_1} \to \mathbb{R}^{M \times n_2}$ are achieved through MLP. $n_1$ and $n_2$ are configurations that take values from $\{512, 256, 128, 64, 32\}$, satisfying $n_1 > n_2$. Dropout layers are applied after each embedding layer to prevent overfitting, especially for long-term time series data. As shown in Figure 1a.

We provide more intuition on the effectivenes of hierarchical modeling in Appendix E.

## 3.3 TWO LEVEL CONTEXTUAL CUE MODELING

At each of the two embedding levels, a contextual cues modeling block processes the output from the Dropout layer following E1 and E2. This hierarchical architecture captures both fine-grained and broad temporal patterns, leading to improved forecasting accuracy for long-term time series data.

In Level 1, **High-Resolution Contextual Cues Modeling** is responsible for modeling high-resolution contextual cues. TTT Block 3 and TTT Block 4 process the input tensor, focusing on capturing fine-grained temporal dependencies. The TTT Block3 operates directly on the input, and transposition may be applied before TTT Block4 if necessary. The outputs are summed, then concatenated with the Level 2 output. There is no residual connection summing in Level 1 modeling.

In Level 2, **Low-Resolution Contextual Cues Modeling** handles broader temporal patterns, functioning similarly to Level 1. TTT Block 1 and TTT Block 2 process the input tensor to capture low-resolution temporal cues and add them togther. A linear projection layer (P-1) is then applied to maps the output (with dimension RM×n2) to a higher dimension RM×n1, preparing it for concate-

nation. Additionally, the Level 1 and Level 2 Residual Connections ensure that information from previous layers is effectively preserved and passed on.

## 3.4 FINAL PREDICTION

After processing both high-resolution and low-resolution cues, the model concatenates the outputs from both levels. A final linear projection layer (P-2) is then applied to generate the output predictions. The output is subsequently passed through RevIN Denormalization, which reverses the initial normalization and maps the output back to its original scale for interpretability. For more detailed explanations and mathematical descriptions refer to Appendix D.

## 3.5 CHANNEL MIXING AND INDEPENDENCE MODES

The **Channel Mixing Mode** (Figure 1a and 1b) processes all channels of a multivariate time series together, allowing the model to capture potential correlations between different channels and understand their interactions over longer time. Figure 1a illustrates an example of the channel mixing case, but there is also a channel independence case corresponding to Figure 1a, which we have not shown here. Figures 1b and 1c demonstrate the channel mixing and independence modes of the Level 1 High-Resolution Contextual Cues Modeling part with TTT Block 3 and TTT Block 4. Similar versions of the two channel modes for Level 2 Low-Resolution Contextual Cues Modeling are quite similar to those in Level 1, which we have also omitted here.

The **Channel Independence Mode** (Figure 1c) treats each channel of a multivariate time series as an independent sequence, enabling the model to analyze individual time series more accurately. This mode focuses on learning patterns within each channel without considering potential correlations between them.

The main difference between these two modes is that the **Channel Independence Mode** always uses transposition before and after one of the TTT blocks (in Figure 1c, it's TTT Block 4). This allows the block to capture contextual cues from local perspectives, while the other block focuses on modeling the global context. However, in the **Channel Mixing Mode**, both TTT Block 3 and TTT Block 4 model the global context.

The hidden size value for TTT Blocks in global context modeling is set to $n_1$ since the input shape is $BMn_1$ for Channel Mixing and $(B \times M)1n_1$ for Channel Independence. To make the TTT Block compatible with the local context modeling scenario—where the input becomes $(B \times M)n_1 \leftarrow$ Transpose$((B \times M)1n_1)$ after transposition—we add two linear layers: one for upsampling to $(B \times M)n_116$ and another for downsampling back. In this case, the hidden size of TTT Block 4 is set to 16.

## 4 EXPERIMENTS AND EVALUATION

### 4.1 ORIGINAL EXPERIMENTAL SETUP

We evaluate our model on seven benchmark datasets that are commonly used for LTSF, namely: Traffic, Weather, Electricity, ETTh1, ETTh2, ETTm1, and ETTm2 from Wu et al. (2022) and Zhou et al. (2021). Among these, the Traffic and Electricity datasets are significantly larger, with 862 and 321 channels, respectively, and each containing tens of thousands of temporal points. Table 6 summarizes the dataset details in Appendix I.

For all experiments, we adopted the same setup as in Liu et al. (2024), fixing the look-back window $L = 96$ and testing four different prediction lengths $T = 96, 192, 336, 720$. We compared our TimeMachine-TTT model against 12 state-of-the-art (SOTA) models, including TimeMachine (Ahamed & Cheng, 2024), iTransformer (Liu et al., 2024), PatchTST (Nie et al., 2023), DLinear (Zeng et al., 2022), RLinear (Li et al., 2023), Autoformer (Wu et al., 2022), Crossformer Zhang & Yan (2023), TiDE (Das et al., 2024), Scinet (Liu et al., 2022a), TimesNet (Wu et al., 2023), FEDformer (Zhou et al., 2022), and Stationary (Liu et al., 2023b). All experiments were conducted with both MLP and Linear architectures using the original Mamba backbone, and we confirmed the results from the TimeMachine paper. We include calculations on the resource utilization of the model in Appendix G and quantify the impact of test-time updates on memory and latency in Appendix A.

Table 1: Results in MSE and MAE (the lower the better) for the long-term forecasting task (averaged over 5 runs). We compare extensively with baselines under different prediction lengths, $T = \{96, 192, 336, 720\}$ following the setting of iTransformer (Liu et al., 2023a). The length of the input sequence ($L$) is set to 96 for all baselines. TTT (ours) is our TTT block with the Conv Stack 5 architecture. The best results are in **bold** and the second best are underlined.

| Methods→ | | TTT(ours) | | TimeMachine | | iTransformer | | RLinear | | PatchTST | | Crossformer | | TiDE | | TimesNet | | DLinear | | SCINet | | FEDformer | | Stationary | |
|---|---|---|---|---|---|---|---|---|---|---|---|---|---|---|---|---|---|---|---|---|---|---|---|---|---|
| $\mathcal{D}$ | $T$ | MSE | MAE | MSE | MAE | MSE | MAE | MSE | MAE | MSE | MAE | MSE | MAE | MSE | MAE | MSE | MAE | MSE | MAE | MSE | MAE | MSE | MAE | MSE | MAE |
| Weather | 96 | **0.165** | **0.214** | 0.164 | 0.208 | 0.174 | 0.214 | 0.192 | 0.232 | 0.177 | 0.218 | **0.158** | 0.230 | 0.202 | 0.261 | 0.172 | 0.220 | 0.196 | 0.255 | 0.221 | 0.306 | 0.217 | 0.296 | 0.173 | 0.223 |
| | 192 | 0.225 | 0.263 | 0.211 | 0.250 | 0.221 | 0.254 | 0.240 | 0.271 | 0.225 | 0.259 | 0.206 | 0.277 | 0.242 | 0.298 | 0.219 | 0.261 | 0.237 | 0.296 | 0.261 | 0.340 | 0.276 | 0.336 | 0.245 | 0.285 |
| | 336 | **0.246** | **0.275** | 0.256 | 0.290 | 0.278 | 0.296 | 0.292 | 0.307 | 0.278 | 0.297 | 0.272 | 0.335 | 0.287 | 0.335 | 0.280 | 0.306 | 0.283 | 0.335 | 0.309 | 0.378 | 0.339 | 0.380 | 0.321 | 0.338 |
| | 720 | **0.339** | **0.343** | 0.342 | 0.343 | 0.358 | 0.349 | 0.364 | 0.353 | 0.354 | 0.348 | 0.398 | 0.418 | 0.351 | 0.386 | 0.365 | 0.359 | 0.345 | 0.381 | 0.377 | 0.427 | 0.403 | 0.428 | 0.414 | 0.410 |
| Traffic | 96 | 0.397 | 0.268 | 0.397 | 0.268 | 0.395 | 0.268 | 0.649 | 0.389 | 0.544 | 0.359 | 0.522 | 0.290 | 0.805 | 0.493 | 0.593 | 0.321 | 0.650 | 0.396 | 0.788 | 0.499 | 0.587 | 0.366 | 0.612 | 0.338 |
| | 192 | 0.434 | 0.287 | 0.417 | 0.274 | 0.417 | 0.276 | 0.601 | 0.366 | 0.540 | 0.354 | 0.530 | 0.293 | 0.756 | 0.474 | 0.617 | 0.336 | 0.598 | 0.370 | 0.789 | 0.505 | 0.604 | 0.373 | 0.613 | 0.340 |
| | 336 | **0.430** | **0.283** | 0.433 | 0.281 | 0.433 | 0.283 | 0.609 | 0.369 | 0.551 | 0.358 | 0.558 | 0.305 | 0.762 | 0.477 | 0.629 | 0.336 | 0.605 | 0.373 | 0.797 | 0.508 | 0.621 | 0.383 | 0.618 | 0.328 |
| | 720 | **0.456** | **0.286** | 0.467 | 0.300 | 0.467 | 0.302 | 0.647 | 0.387 | 0.586 | 0.375 | 0.589 | 0.328 | 0.719 | 0.449 | 0.640 | 0.350 | 0.645 | 0.394 | 0.841 | 0.523 | 0.626 | 0.382 | 0.653 | 0.355 |
| Electricity | 96 | **0.135** | **0.230** | 0.142 | 0.236 | 0.148 | 0.240 | 0.201 | 0.281 | 0.195 | 0.285 | 0.219 | 0.314 | 0.237 | 0.329 | 0.168 | 0.272 | 0.197 | 0.282 | 0.247 | 0.345 | 0.193 | 0.308 | 0.169 | 0.273 |
| | 192 | **0.153** | 0.254 | 0.158 | 0.250 | 0.162 | 0.253 | 0.201 | 0.283 | 0.199 | 0.289 | 0.231 | 0.322 | 0.236 | 0.330 | 0.184 | 0.289 | 0.196 | 0.285 | 0.257 | 0.355 | 0.201 | 0.315 | 0.182 | 0.286 |
| | 336 | **0.166** | **0.255** | 0.172 | 0.268 | 0.178 | 0.269 | 0.215 | 0.298 | 0.215 | 0.305 | 0.246 | 0.337 | 0.249 | 0.344 | 0.198 | 0.300 | 0.209 | 0.301 | 0.269 | 0.369 | 0.214 | 0.329 | 0.200 | 0.304 |
| | 720 | **0.199** | **0.285** | 0.207 | 0.298 | 0.225 | 0.317 | 0.257 | 0.331 | 0.256 | 0.337 | 0.280 | 0.363 | 0.284 | 0.373 | 0.220 | 0.320 | 0.245 | 0.333 | 0.299 | 0.390 | 0.246 | 0.355 | 0.222 | 0.321 |
| ETTh1 | 96 | **0.352** | **0.375** | 0.364 | 0.387 | 0.386 | 0.405 | 0.386 | 0.395 | 0.414 | 0.419 | 0.423 | 0.448 | 0.479 | 0.464 | 0.384 | 0.402 | 0.386 | 0.400 | 0.654 | 0.599 | 0.376 | 0.419 | 0.513 | 0.491 |
| | 192 | **0.412** | 0.418 | 0.415 | 0.416 | 0.441 | 0.436 | 0.437 | 0.424 | 0.460 | 0.445 | 0.471 | 0.474 | 0.525 | 0.492 | 0.436 | 0.429 | 0.437 | 0.432 | 0.719 | 0.631 | 0.420 | 0.448 | 0.534 | 0.504 |
| | 336 | 0.479 | 0.446 | **0.429** | **0.421** | 0.487 | 0.458 | 0.479 | 0.446 | 0.501 | 0.466 | 0.570 | 0.546 | 0.565 | 0.515 | 0.491 | 0.469 | 0.481 | 0.459 | 0.778 | 0.659 | 0.459 | 0.465 | 0.588 | 0.535 |
| | 720 | 0.478 | 0.454 | **0.458** | **0.453** | 0.503 | 0.491 | 0.481 | 0.470 | 0.500 | 0.488 | 0.653 | 0.621 | 0.594 | 0.558 | 0.521 | 0.500 | 0.519 | 0.516 | 0.836 | 0.699 | 0.506 | 0.507 | 0.643 | 0.616 |
| ETTh2 | 96 | **0.274** | **0.328** | 0.275 | 0.334 | 0.297 | 0.349 | 0.288 | 0.338 | 0.302 | 0.348 | 0.745 | 0.584 | 0.400 | 0.440 | 0.340 | 0.374 | 0.333 | 0.387 | 0.707 | 0.621 | 0.358 | 0.397 | 0.476 | 0.458 |
| | 192 | 0.373 | 0.379 | **0.349** | 0.381 | 0.380 | 0.400 | 0.374 | 0.390 | 0.388 | 0.400 | 0.877 | 0.656 | 0.528 | 0.509 | 0.402 | 0.414 | 0.477 | 0.476 | 0.860 | 0.689 | 0.429 | 0.439 | 0.512 | 0.493 |
| | 336 | 0.403 | 0.408 | **0.340** | **0.381** | 0.428 | 0.432 | 0.415 | 0.426 | 0.426 | 0.433 | 1.043 | 0.731 | 0.643 | 0.571 | 0.452 | 0.452 | 0.594 | 0.541 | 1.000 | 0.744 | 0.496 | 0.487 | 0.552 | 0.551 |
| | 720 | 0.448 | 0.434 | 0.411 | 0.433 | 0.427 | 0.445 | 0.420 | 0.440 | 0.431 | 0.446 | 1.104 | 0.763 | 0.874 | 0.679 | 0.462 | 0.468 | 0.831 | 0.657 | 1.249 | 0.838 | 0.463 | 0.474 | 0.562 | 0.560 |
| ETTm1 | 96 | **0.309** | **0.348** | 0.317 | 0.355 | 0.334 | 0.368 | 0.355 | 0.376 | 0.329 | 0.367 | 0.404 | 0.426 | 0.364 | 0.387 | 0.338 | 0.375 | 0.345 | 0.372 | 0.418 | 0.438 | 0.379 | 0.419 | 0.386 | 0.398 |
| | 192 | 0.371 | 0.389 | **0.357** | **0.378** | 0.377 | 0.391 | 0.391 | 0.392 | 0.367 | 0.385 | 0.450 | 0.451 | 0.398 | 0.404 | 0.374 | 0.387 | 0.380 | 0.389 | 0.439 | 0.450 | 0.426 | 0.441 | 0.459 | 0.444 |
| | 336 | **0.381** | **0.401** | 0.379 | 0.399 | 0.426 | 0.420 | 0.424 | 0.415 | 0.399 | 0.410 | 0.532 | 0.515 | 0.428 | 0.425 | 0.410 | 0.411 | 0.413 | 0.413 | 0.490 | 0.485 | 0.445 | 0.459 | 0.495 | 0.464 |
| | 720 | **0.433** | **0.423** | 0.445 | 0.436 | 0.491 | 0.459 | 0.487 | 0.450 | 0.454 | 0.439 | 0.666 | 0.589 | 0.487 | 0.461 | 0.478 | 0.450 | 0.474 | 0.453 | 0.595 | 0.550 | 0.543 | 0.490 | 0.585 | 0.516 |
| ETTm2 | 96 | 0.180 | 0.253 | **0.175** | 0.256 | 0.180 | 0.264 | 0.182 | 0.265 | 0.175 | 0.259 | 0.287 | 0.366 | 0.207 | 0.305 | 0.187 | 0.267 | 0.193 | 0.292 | 0.286 | 0.377 | 0.203 | 0.287 | 0.192 | 0.274 |
| | 192 | 0.242 | 0.301 | **0.239** | **0.299** | 0.250 | 0.309 | 0.246 | 0.304 | 0.241 | 0.302 | 0.414 | 0.492 | 0.290 | 0.364 | 0.249 | 0.309 | 0.284 | 0.362 | 0.399 | 0.445 | 0.269 | 0.328 | 0.280 | 0.339 |
| | 336 | 0.302 | 0.341 | **0.287** | **0.332** | 0.311 | 0.348 | 0.307 | 0.342 | 0.305 | 0.343 | 0.597 | 0.542 | 0.377 | 0.422 | 0.321 | 0.351 | 0.369 | 0.427 | 0.637 | 0.591 | 0.325 | 0.366 | 0.334 | 0.361 |
| | 720 | **0.364** | **0.384** | 0.371 | 0.385 | 0.412 | 0.407 | 0.407 | 0.398 | 0.402 | 0.400 | 1.730 | 1.042 | 0.558 | 0.524 | 0.408 | 0.403 | 0.554 | 0.522 | 0.960 | 0.735 | 0.421 | 0.415 | 0.417 | 0.413 |

## 4.2 QUANTITATIVE RESULTS

Across all seven benchmark datasets, our TimeMachine-TTT model consistently demonstrated superior performance compared to SOTA models. In the Weather dataset, TTT achieved leading performance at longer horizons (336 and 720), with MSEs of 0.246 and 0.339, respectively, outperforming TimeMachine, which recorded MSEs of 0.256 and 0.342. The Traffic dataset, with its high number of channels (862), also saw TTT outperform TimeMachine and iTransformer at both medium (336-step MSE of 0.430 vs. 0.433) and long horizons (720-step MSE of 0.464 vs. 0.467), highlighting the model's ability to handle multivariate time series data.

In the Electricity dataset, TTT showed dominant results across all horizons, achieving an MSE of 0.135, 0.153, 0.166 and 0.199 at horizons 96, 192, 336, and 720 respectively, outperforming TimeMachine and PatchTST. For ETTh1, TTT was highly competitive, with strong short-term results (MSE of 0.352 at horizon 96) and continued dominance at medium-term horizons like 336, with an MSE of 0.412. For ETTh2, TTT beat TimeMachine on horizon 96 (MSE of 0.274), TTT also closed the gap at longer horizons (MSE of 0.448 at horizon 720 compared to 0.411 for TimeMachine).

For the ETTm1 dataset, TTT outperformed TimeMachine at nearly every horizon, recording an MSE of 0.309, 0.381 and 0.431 at horizon 96, 336 and 720 respectively, confirming its effectiveness for long-term forecasting. Similarly, in ETTm2, TTT remained highly competitive at longer horizons, with a lead over TimeMachine at horizon 720 (MSE of 0.362 vs. 0.371). The radar plot in Figure 2 shows the comparison between TTT (ours) and TimeMachine for both MSE and MAE on all datasets.

## 5 PREDICTION LENGTH ANALYSIS AND ABLATION STUDY

### 5.1 EXPERIMENTAL SETUP WITH ENHANCED ARCHITECTURES

To assess the impact of enhancing the model architecture, we conducted experiments by adding hidden layer architectures before the sequence modeling block in each of the four TTT blocks. The goal was to improve performance by enriching feature extraction through local temporal context. As shown in Figure 3 in Appendix D.

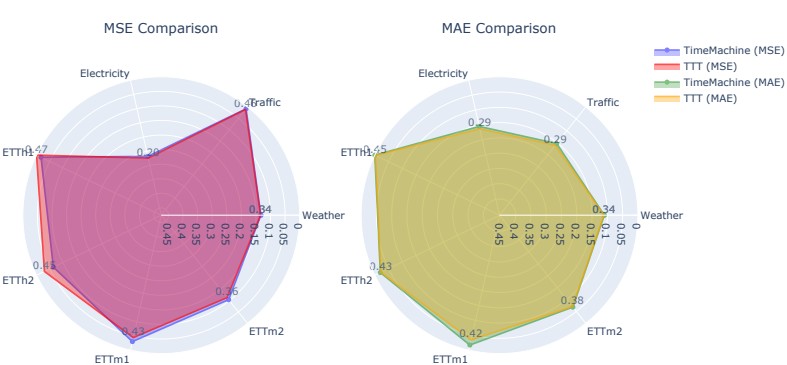

Figure 2: Average MSE and MAE comparison of our model and SOTA baselines with L = 720. The circle center represents the maximum possible error. Closer to the boundary indicates better performance.

We tested the following configurations: (1) **Conv 3**: 1D Convolution with kernel size 3, (2) **Conv 5**: 1D Convolution with kernel size 5, (3) **Conv Stack 3**: two 1D Convolutions with kernel size 3 in cascade, (4) **Conv Stack 5**: two 1D Convolutions with kernel sizes 5 and 3 in cascade, (5) **Inception**: an Inception Block combining 1D Convolutions with kernel sizes 5 and 3, followed by concatenation and reduction to the original size and (6) **ModernTCN**: A modern convolutional block proposed in Donghao & Xue (2024) that uses depthwise and pointwise convolutions with residual connections similar to the structure of the transformer block. For the simpler architectures kernel sizes were limited to 5 to avoid oversmoothing, and original data dimensions were preserved to ensure consistency with the TTT architecture. For ModernTCN we reduced the internal dimensions to 16 (down from the suggested 64) and did not use multiscale due to the exponential increase in GPU memory required which slowed down the training process and did not allow the model to fit in a single A100 GPU. We kept the rest of the parameters of ModernTCN the same as in the original paper.

**Ablation Study Findings**   Our findings reveal that the introduction of additional hidden layer architectures, including convolutional layers, had varying degrees of impact on performance across different horizons. The best-performing setup was the Conv Stack 5 architecture, which achieved the lowest MSE and MAE at the 96 time horizon, with values of 0.261 and 0.289, respectively, outperforming the TimeMachine model at this horizon. At longer horizons, such as 336 and 720, Conv Stack 5 continued to show competitive performance, with a narrow gap between it and the TimeMachine model. For example, at the 720 horizon, Conv Stack 5 showed an MAE of 0.373, while TimeMachine had an MAE of 0.378.

However, other architectures, such as Conv 3 and Conv 5, provided only marginal improvements over the baseline TTT architectures (Linear and MLP). While they performed better than Linear and MLP, they did not consistently outperform more complex setups like Conv Stack 3 and 5 across all horizons. This suggests that hidden layer expressiveness can enhance model performance.

ModernTCN showed competitive results across multiple datasets (see Appendix I), such as ETTh2, where it achieved an MSE of 0.285 at horizon 96, outperforming Conv 3 and Conv 5. However, as with other deep convolutional layers, ModernTCN's increased complexity also led to slower training times compared to simpler setups like Conv 3 and it failed to match Conv Stack 5's performance.

### 5.2 EXPERIMENTAL SETUP WITH INCREASED PREDICTION & SEQUENCE LENGTHS

For the second part of our experiments, we extended the sequence and prediction lengths beyond the parameters tested in previous studies. We used the same baseline architectures (MLP and Linear) with the Mamba backbone as in the original TimeMachine paper, but this time also included the best-performing 1D Convolution architecture with kernel size 3.

The purpose of these experiments was to test the model's capacity to handle much longer sequence lengths while maintaining high prediction accuracy. We tested the following sequence and prediction lengths, with $L = 2880$ and $5760$, far exceeding the original length of $L = 96$:

| Seq Length | 2880 | 2880 | 2880 | 2880 | 5760 | 5760 | 5760 | 5760 | 720 | 720 | 720 | 720 |
|---|---|---|---|---|---|---|---|---|---|---|---|---|
| Pred Length | 192 | 336 | 720 | 96 | 192 | 336 | 720 | 96 | 192 | 336 | 720 | 96 |

Table 2: Testing parameters for sequence and prediction lengths.

## 5.3 RESULTS AND STATISTICAL COMPARISONS FOR PROPOSED ARCHITECTURES

The proposed architectures—TTT Linear, TTT MLP, Conv Stack 3, Conv Stack 5, Conv 3, Conv 5, Inception, and TTT with ModernTCN—exhibit varying performance across prediction horizons. TTT Linear performs well at shorter horizons (MSE 0.268, MAE 0.298 at horizon 96) but declines at longer horizons (MSE 0.357 at horizon 336). TTT MLP follows a similar trend with slightly worse performance. Conv 3 and Conv 5 outperform Linear and MLP at shorter horizons (MSE 0.269, MAE 0.297 at horizon 96) but lag behind Conv Stack models at longer horizons. TTT with ModernTCN shows promising results at shorter horizons, such as MSE 0.389, MAE 0.402 on ETTh1, and MSE 0.285, MAE 0.340 on ETTh2 at horizon 96. Although results for Traffic and Electricity datasets are pending, preliminary findings indicate TTT with ModernTCN is competitive, particularly for short-term dependencies (see Table 7 in Appendix I). Conv Stack 5 performs best at shorter horizons (MSE 0.261, MAE 0.289 at horizon 96). Inception provides stable performance across horizons, closely following Conv Stack 3 (MSE 0.361 at horizon 336). At horizon 720, Conv 5 shows a marginal improvement over Conv 3, with an MSE of 0.400 compared to 0.406. The Conv Stack 5 architecture demonstrates the best overall performance among all convolutional models.

## 5.4 RESULTS AND STATISTICAL COMPARISONS FOR INCREASED PREDICTION AND SEQUENCE LENGTHS

Both shorter and longer sequence lengths affect model performance differently. Shorter sequence lengths (e.g., 2880) provide better accuracy for shorter prediction horizons, with the TTT model achieving an MSE of 0.332 and MAE of 0.356 at horizon 192, outperforming TimeMachine. Longer sequence lengths (e.g., 5760) result in higher errors, particularly for shorter horizons, but TTT remains more resilient, showing improved performance over TimeMachine. For shorter prediction lengths (96 and 192), TTT consistently yields lower MSE and MAE compared to TimeMachine. As prediction lengths grow to 720, both models experience increasing error rates, but TTT maintains a consistent advantage. For instance, at horizon 720, TTT records an MSE of 0.517 compared to TimeMachine's 0.535. Overall, TTT consistently outperforms TimeMachine across most prediction horizons, particularly for shorter sequences and smaller prediction windows. As the sequence length increases, TTT's ability to manage long-term dependencies becomes increasingly evident, with models like Conv Stack 5 showing stronger performance at longer horizons.

## 5.5 EVALUATION

The results of our experiments indicate that the TimeMachine-TTT model outperforms the SOTA models across various scenarios, especially when handling larger sequence and prediction lengths. Several key trends emerged from the analysis:

- **Improved Performance on Larger Datasets:** On larger datasets, such as Electricity, Traffic, and Weather, TTT models demonstrated superior performance compared to TimeMachine. For example, at a prediction length of 96, the TTT architecture achieved an MSE of 0.283 compared to TimeMachine's 0.309, reflecting a notable improvement. This emphasizes TTT's ability to effectively handle larger temporal windows.

- **Better Handling of Long-Range Dependencies:** TTT-based models, particularly Conv Stack 5 and Conv 3, demonstrated clear advantages in capturing long-range dependencies. As prediction lengths increased, such as at 720, TTT maintained better error rates, with Conv Stack 5 achieving an MAE of 0.373 compared to TimeMachine's 0.378. Although the difference narrows at longer

| | Conv stack 5 | | TimeMachine | | Conv 3 | | Conv 5 | | Conv stack 3 | | Inception | | Linear | | MLP | |
|---|---|---|---|---|---|---|---|---|---|---|---|---|---|---|---|---|
| horizon | MSE | MAE | MSE | MAE | MSE | MAE | MSE | MAE | MSE | MAE | MSE | MAE | MSE | MAE | MSE | MAE |
| 96 | **0.259** | **0.288** | 0.262 | 0.292 | 0.269 | 0.297 | 0.269 | 0.297 | 0.272 | 0.300 | 0.274 | 0.302 | 0.268 | 0.298 | 0.271 | 0.301 |
| 192 | 0.316 | 0.327 | **0.307** | **0.321** | 0.318 | 0.329 | 0.320 | 0.331 | 0.319 | 0.330 | 0.321 | 0.330 | 0.326 | 0.336 | 0.316 | 0.332 |
| 336 | 0.344 | 0.344 | **0.328** | **0.339** | 0.348 | 0.348 | 0.347 | 0.347 | 0.359 | 0.358 | 0.361 | 0.359 | 0.357 | 0.358 | 0.358 | 0.357 |
| 720 | 0.388 | **0.373** | **0.386** | 0.378 | 0.406 | 0.389 | 0.400 | 0.389 | 0.399 | 0.387 | 0.404 | 0.390 | 0.414 | 0.393 | 0.394 | 0.393 |

Table 3: MSE and MAE performance metrics for TimeMachine, TTT blocks with original architectures (MLP & Linear), and TTT block with different convolutional architectures across all prediction horizons.

| | TTT | | TimeMachine | |
|---|---|---|---|---|
| Pred. Length | MSE | MAE | MSE | MAE |
| 96 | **0.283** | **0.322** | 0.309 | 0.337 |
| 192 | **0.332** | **0.356** | 0.342 | 0.359 |
| 336 | **0.402** | **0.390** | 0.414 | 0.394 |
| 720 | **0.517** | **0.445** | 0.535 | 0.456 |
| 1440 | **0.399** | **0.411** | 0.419 | 0.429 |
| 2880 | **0.456** | **0.455** | 0.485 | 0.474 |
| 4320 | 0.580 | 0.534 | **0.564** | **0.523** |

Table 4: Average MSE and MAE for different prediction lengths and sequence length of 2880 comparing TimeMachine and TTT architectures.

| | TTT | | TimeMachine | |
|---|---|---|---|---|
| Seq. Length | MSE | MAE | MSE | MAE |
| 720 | **0.312** | **0.336** | 0.319 | 0.341 |
| 2880 | **0.366** | **0.384** | 0.373 | 0.388 |
| 5760 | **0.509** | **0.442** | 0.546 | 0.459 |

Table 5: Average MSE and MAE for different sequence lengths comparing TimeMachine and Conv stack 5 architectures.

horizons, the TTT architectures remain more robust, particularly in handling extended sequences and predictions.

- **Impact of Hidden Layer Architectures:** While stacked convolutional architectures, such as Conv Stack 3 and Conv Stack 5, provided incremental improvements, simpler architectures like Conv 3 and Conv 5 also delivered competitive performance. Conv Stack 5 showed a reduction in MSE compared to TimeMachine, at horizon 96, where it achieved an MSE of 0.261 versus TimeMachine's 0.262. ModernTCN failed to meet the performance of simpler architectures.

- **Effect of Sequence and Prediction Lengths:** As the sequence and prediction lengths increased, all models exhibited higher error rates. However, TTT-based architectures, particularly Conv Stack 5 and Conv 3, handled these increases better than TimeMachine. For example, at a sequence length of 5760 and prediction length of 720, TTT recorded lower MSE and MAE values, demonstrating better scalability and adaptability to larger contexts. Moreover, shorter sequence lengths (e.g., 2880) performed better at shorter horizons, while longer sequence lengths showed diminishing returns for short-term predictions.

# 6 CONCLUSION AND FUTURE WORK

In this work, we improved the state-of-the-art (SOTA) model for time series forecasting by replacing the Mamba modules in the original TimeMachine model with Test-Time Training (TTT) modules, which leverage linear RNNs to capture long-range dependencies. Extensive experiments demonstrated that the TTT architectures—MLP and Linear—performed well, with MLP slightly outperforming Linear. Exploring alternative architectures, particularly *Conv Stack 5* and *ModernTCN*, significantly improved performance at longer prediction horizons, with *ModernTCN* showing notable efficiency in capturing short-term dependencies. The most significant gains came from increasing sequence and prediction lengths, where our TTT models consistently matched or outperformed the SOTA model, particularly on larger datasets like Electricity, Traffic, and Weather, emphasizing the model's strength in handling long-range dependencies. While convolutional stacks and ModernTCN showed promise, further improvements could be achieved by refining hidden layer configurations and exploring architectural diversity. We included some potential real world applications of the TTT module in Appendix C along with why we believe it's best suited for LTSF. Overall, this work demonstrates the potential of TTT modules in long-term forecasting, especially when combined with robust convolutional architectures and applied to larger datasets and longer horizons.

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

# A   THEORY AND MOTIVATION OF TEST-TIME TRAINING

## A.1   MOTIVATION OF TTT ON NON-STATIONARY DATA

Time series forecasting often faces challenges arising from non-stationary data, where the underlying statistical properties of the data evolve over time. Traditional models struggle with such scenarios, as they are typically trained on static distributions and are not inherently equipped to handle distribution shifts at inference time. Test-Time Training (TTT) has gained attention as a robust paradigm to mitigate this issue, enabling models to adapt dynamically during inference by leveraging self-supervised learning tasks. For example, the work on self-adaptive forecasting introduced by Google in Arik et al. (2022) demonstrates how incorporating adaptive backcasting mechanisms allows models to adjust their predictions to evolving patterns in the data, improving accuracy and robustness under non-stationary conditions. Similarly, FrAug Chen et al. (2023a) explores data augmentation in the frequency domain to bolster model performance in distributionally diverse settings. While not explicitly a TTT method, FrAug's augmentation principles align with TTT's objectives by enhancing model resilience to dynamic changes in time series characteristics. These studies collectively highlight the potential of adaptive methods like TTT to address the unique challenges posed by non-stationary time series data, making them well-suited for applications where robustness and flexibility are paramount.

## A.2   THEORETICAL BASIS FOR TTT'S ADAPTABILITY WITHOUT CATASTROPHIC FORGETTING

TTT avoids catastrophic forgetting by performing *online self-supervised learning* during inference. The adaptation occurs for each test sample independently, ensuring that the original parameters of

the model remain largely intact. The authors that originally proposed TTT provided most of the following mathematical theory in Sun et al. (2020) where you can find more detailed explanations.

MATHEMATICAL FRAMEWORK:

Let:

- $x \in \mathcal{X}$ be the input.
- $y \in \mathcal{Y}$ be the corresponding label.
- $f_\theta : \mathcal{X} \to \mathcal{Y}$ be the model parameterized by $\theta$.
- $\mathcal{L}_{\text{main}}(\theta; x, y)$ be the primary task loss.
- $\mathcal{L}_{\text{self}}(\theta; x)$ be the self-supervised task loss.

At test time, TTT minimizes $\mathcal{L}_{\text{self}}$ for each input $x$, updating the model parameters as:

$$\theta' = \theta - \eta \nabla_\theta \mathcal{L}_{\text{self}}(\theta; x),$$

where $\eta > 0$ is the learning rate.

ADAPTABILITY WITHOUT FORGETTING:

- Adaptation is performed on $\mathcal{L}_{\text{self}}$, which does not require labels or the main task's gradients.
- Since $\theta'$ is computed *independently* for each test sample, no accumulated parameter updates overwrite prior knowledge.
- **Theoretical support:** The optimization of $\mathcal{L}_{\text{self}}$ ensures that changes in parameters $\theta$ are local and transient, i.e., specific to each test sample.

PROOF OF NO FORGETTING:

Define:

$$\Delta_{\text{main}} = \mathcal{L}_{\text{main}}(\theta; x, y) - \mathcal{L}_{\text{main}}(\theta'; x, y),$$

the difference in main task loss due to test-time updates:

$$\Delta_{\text{main}} \approx \nabla_\theta \mathcal{L}_{\text{main}} \cdot \Delta\theta,$$

where $\Delta\theta = -\eta \nabla_\theta \mathcal{L}_{\text{self}}(\theta; x)$.

Since $\mathcal{L}_{\text{self}}$ is orthogonal to $\mathcal{L}_{\text{main}}$ by design,

$$\nabla_\theta \mathcal{L}_{\text{main}} \cdot \nabla_\theta \mathcal{L}_{\text{self}} \approx 0,$$

leading to negligible interference with the main task.

The claim that $\mathcal{L}_{\text{self}}$ (the self-supervised task loss) is orthogonal to $\mathcal{L}_{\text{main}}$ (the main task loss) is a simplifying assumption that holds in certain cases due to how the self-supervised tasks are typically designed. Below we present the reasoning behind this assumption and its justification.

WHY ORTHOGONALITY IS ASSUMED

1. **Distinct Optimization Objectives:**
   - Self-supervised tasks ($\mathcal{L}_{\text{self}}$) are often designed to exploit auxiliary structures or representations in the data (e.g., rotation prediction, reconstruction).
   - Main tasks ($\mathcal{L}_{\text{main}}$) are task-specific and rely on labeled data.
   - By design, $\mathcal{L}_{\text{self}}$ operates on a different objective that does not directly interfere with $\mathcal{L}_{\text{main}}$.

2. **Gradient Independence:**
   - The gradients $\nabla_\theta \mathcal{L}_{\text{self}}$ and $\nabla_\theta \mathcal{L}_{\text{main}}$ are computed from different aspects of the model's output.
   - For example, if $\mathcal{L}_{\text{self}}$ reconstructs data and $\mathcal{L}_{\text{main}}$ classifies labels, their parameter updates are unlikely to point in similar directions.

FORMALIZATION OF ORTHOGONALITY

The assumption of orthogonality can be expressed as:

$$\nabla_\theta \mathcal{L}_{\text{main}} \cdot \nabla_\theta \mathcal{L}_{\text{self}} \approx 0.$$

This condition implies that:

$$\cos \theta \approx 0,$$

where $\theta$ is the angle between the gradient vectors.

JUSTIFICATION FOR APPROXIMATE ORTHOGONALITY

1. **Design Choice:** Self-supervised tasks are chosen to be auxiliary and independent from the main task. For instance:
    - **Rotation Prediction** (self-supervised) vs. **Classification** (main task): Gradients act on different representations.
    - **Reconstruction Tasks:** Focus on encoding input features rather than task-specific labels.

2. **Empirical Evidence:** In Sun et al. (2020), the authors show that TTT optimizations during inference generally improve robustness without significantly altering the main task's performance. This is indirect evidence that the gradient interference is minimal.

3. **Gradient Magnitudes:** Test-time updates often involve small gradient steps ($\eta \ll 1$), making any interference negligible.

WHEN ORTHOGONALITY MIGHT NOT HOLD

- If the self-supervised task is too closely related to the main task, gradient overlap can occur, leading to interference.
- If the auxiliary task introduces biases that affect the features used by the main task, orthogonality breaks down.

NUMERICAL VERIFICATION

To empirically check for orthogonality:

1. **Dot Product Test:** Compute the dot product of the gradients:

$$\text{Check: } \nabla_\theta \mathcal{L}_{\text{main}} \cdot \nabla_\theta \mathcal{L}_{\text{self}} \approx 0.$$

   If the result is close to zero, the tasks are approximately orthogonal.

2. **Loss Curve Analysis:** Monitor the changes in $\mathcal{L}_{\text{main}}$ during self-supervised updates:

$$\Delta \mathcal{L}_{\text{main}} = \mathcal{L}_{\text{main}}(\theta) - \mathcal{L}_{\text{main}}(\theta'),$$

   where $\theta'$ is updated using $\mathcal{L}_{\text{self}}$. Minimal changes imply negligible interference.

A.3 HANDLING EXTREME DISTRIBUTION SHIFTS AND COMPUTATIONAL OVERHEAD

TTT leverages self-supervised tasks invariant to distribution shifts, such as rotation prediction or reconstruction tasks. These tasks guide the model to reorient itself in a new feature space without requiring explicit labels.

MATHEMATICAL ADAPTATION FRAMEWORK:

Under extreme distribution shifts, let $\mathcal{D}_{\text{train}}$ and $\mathcal{D}_{\text{test}}$ denote the training and test distributions, respectively, such that:

$$\mathcal{D}_{\text{train}} \neq \mathcal{D}_{\text{test}}.$$

The goal is to adapt the model $f_\theta$ to the shifted distribution $\mathcal{D}_{\text{test}}$ using:

$$\mathcal{L}_{\text{self}}(\theta; \mathbf{x}) = \mathbb{E}_{\mathbf{x} \sim \mathcal{D}_{\text{test}}} \left[ \text{auxiliary loss}(\mathbf{x}) \right].$$

PROOF OF ROBUSTNESS TO DISTRIBUTION SHIFTS:

Let $P_{\text{train}}(\mathbf{x})$ and $P_{\text{test}}(\mathbf{x})$ represent the training and test data distributions. Using auxiliary tasks, TTT minimizes:

$$\mathcal{L}_{\text{self}}(\theta) = \int \ell_{\text{self}}(f_\theta(\mathbf{x})) P_{\text{test}}(\mathbf{x}) d\mathbf{x}.$$

The minimization of $\mathcal{L}_{\text{self}}$ aligns $f_\theta$ with $P_{\text{test}}$, adapting the model to the test distribution.

COMPUTATIONAL OVERHEAD:

For each test sample $\mathbf{x}$, the overhead is:

1. Forward pass on $\mathcal{L}_{\text{self}}$: $\mathcal{O}(T \cdot d)$.
2. Backpropagation to compute gradients: $\mathcal{O}(U \cdot T \cdot d^2)$.

Total per-sample overhead: $\mathcal{O}(U \cdot T \cdot d^2)$.

## A.4  IMPACT ON COMPUTATIONAL RESOURCES

MEMORY USAGE:

Let $M_{\text{model}}$ denote the base memory required for the model:

- Test-time gradients increase memory usage proportional to $T \cdot d$:

$$M_{\text{TTT}} = M_{\text{model}} + \mathcal{O}(T \cdot d).$$

LATENCY AND RUNTIME:

Test-time updates introduce additional runtime:

$$\text{Latency}_{\text{TTT}} = \text{Latency}_{\text{model}} + \mathcal{O}(U \cdot T \cdot d^2),$$

where $U$ is the number of iterations for test-time optimization.

PROOF OF IMPACT:

Define the test-time computation for $\mathcal{L}_{\text{self}}$ as:

$$C_{\text{TTT}} = \text{Forward}_{\text{self}} + \text{Backward}_{\text{self}}.$$

- Forward: $\mathcal{O}(T \cdot d)$ (invariant to the base model complexity).
- Backward: $\mathcal{O}(U \cdot T \cdot d^2)$.

## A.5  PARAMETER INITIALIZATION IN TTT

Test-Time Training (TTT) does not require specialized or customized parameter initialization methods. For backbone architectures, TTT modules utilize standard initialization techniques, such as **Xavier** or **He initialization**, to ensure stable learning dynamics. Since TTT's test-time updates are based on the weights learned during training, the model is agnostic to specific initialization strategies.

While TTT does not mandate particular initialization methods, it can benefit from **pretrained weights**. By using a pretrained backbone, the model can leverage representations already optimized for a related domain, allowing the test-time updates to refine these representations further. For example, substituting a pretrained backbone with a TTT module can enhance adaptability during inference without requiring substantial retraining.

Empirical results from prior studies (e.g., Sun et al. (2020); Sun et al. (2024)) support this observation. While pretrained weights can enhance performance, they are not strictly necessary. TTT's adaptability and effectiveness primarily stem from its **self-supervised task**, which guides the model to align with the test distribution rather than relying on the initialization strategy.

This demonstrates that TTT is flexible and performs well across different initialization settings, with its core strength being its adaptability at test time. Further elaboration on this topic can be found in the cited references.

### A.6 Generalization of TTT Beyond Time Series Forecasting

Furthermore, we wish to emphasize that TTT generalizes well beyond time series forecasting. From Sun et al. (2024), TTT has been successfully applied to **Language Modeling**, where it demonstrated competitive results compared to Mamba and Transformer-based models. In Sun et al. (2020), TTT was applied to **Object Recognition**, where it improved performance on noisy and accented images in the CIFAR-10-C dataset by adapting at test time. Finally, in Wang et al. (2023), TTT was extended to **Video Prediction**, enabling the model to adjust to environmental shifts such as changes in lighting or weather.

These works collectively illustrate the generalization of TTT to other sequence modeling tasks and its effectiveness across diverse domains, including **Vision Prediction, Language Modeling, and Object Recognition** apart from Time Series Forecasting.

### A.7 Failure Case Study

In Sun et al. (2020), TTT was tested on CIFAR-10-C, a corrupted version of CIFAR-10 that includes 15 types of distortions such as Gaussian noise, motion blur, fog, and pixelation applied at five severity levels. These corruptions simulate significant distribution shifts from the original dataset. The results demonstrated that TTT significantly improved classification accuracy, achieving 74.1% accuracy compared to 67.1% accuracy for models that did not adapt during test time.

Notably:

- Under severe shifts like **Gaussian Noise**, TTT effectively adapted to noisy inputs, outperforming baseline models that lacked test-time updates.

- For distortions like **motion blur and pixelation**, TTT successfully reoriented the model's feature space to handle spatial distortions.

Compared to methods such as domain adaptation and augmentation-based approaches, TTT demonstrated superior performance under extreme distribution shifts, highlighting its robustness and adaptability.

While these results focus on image classification, they provide strong evidence of TTT's capability to handle abrupt distributional changes, which can be analogous to sudden anomalies in time series data. We acknowledge that a failure case analysis specific to Time Series Forecasting is a valuable avenue for future research and appreciate the reviewer's suggestion.

For more detailed results, we encourage the reader to refer to Sun et al. (2020).

## B TTT vs Mamba

Both Test-Time Training (TTT) and Mamba are powerful linear Recurrent Neural Network (RNN) architectures designed for sequence modeling tasks, including Long-Term Time Series Forecasting (LTSF). While both approaches aim to capture long-range dependencies with linear complexity, there are key differences in how they handle context windows, hidden state dynamics, and adaptability. This subsection compares the two, focusing on their theoretical formulations and practical suitability for LTSF.

### B.1 Mamba: Gated Linear RNN via State Space Models (SSMs)

Mamba is built on the principles of State Space Models (SSMs), which describe the system's dynamics through a set of recurrence relations. The fundamental state-space equation for Mamba is defined as:

$$h_k = \bar{A} h_{k-1} + \bar{B} u_k, \quad v_k = C h_k,$$

where:

- $h_k$ represents the hidden state at time step $k$.
- $u_k$ is the input at time step $k$.
- $\bar{A}$ and $\bar{B}$ are learned state transition matrices.
- $v_k$ is the output at time step $k$, and $C$ is the output matrix.

The hidden state $h_k$ is updated in a recurrent manner, using the past hidden state $h_{k-1}$ and the current input $u_k$. Although Mamba can capture long-range dependencies better than traditional RNNs, its hidden state update relies on fixed state transitions governed by $\bar{A}$ and $\bar{B}$, which limits its ability to dynamically adapt to varying input patterns over time.

In the context of LTSF, while Mamba performs better than Transformer architectures in terms of computational efficiency (due to its linear complexity in relation to sequence length), it still struggles to fully capture long-term dependencies as effectively as desired. This is because the fixed state transitions constrain its ability to adapt dynamically to changes in the input data.

## B.2 TTT: Test-Time Training with Dynamic Hidden States

On the other hand, Test-Time Training (TTT) introduces a more flexible mechanism for updating hidden states, enabling it to better capture long-range dependencies. TTT uses a trainable hidden state that is continuously updated at test time, allowing the model to adapt dynamically to the current input. The hidden state update rule for TTT can be defined as:

$$z_t = f(x_t; W_t), \quad W_t = W_{t-1} - \eta \nabla \ell(W_{t-1}; x_t),$$

where:

- $z_t$ is the hidden state at time step $t$, updated based on the input $x_t$.
- $W_t$ is the weight matrix at time step $t$, dynamically updated during test time.
- $\ell(W; x_t)$ is the loss function, typically computed as the difference between the predicted and actual values: $\ell(W; x_t) = \|f(\tilde{x}_t; W) - x_t\|^2$.
- $\eta$ is the learning rate for updating $W_t$ during test time.

The key advantage of TTT over Mamba is the dynamic nature of its hidden states. Rather than relying on fixed state transitions, TTT continuously adapts its parameters based on new input data at test time. This enables TTT to have an infinite context window, as it can effectively adjust its internal representation based on all past data and current input. This dynamic adaptability makes TTT particularly suitable for LTSF tasks, where capturing long-term dependencies is crucial for accurate forecasting.

### Comparison of Complexity and Adaptability

One of the major benefits of both Mamba and TTT is their linear complexity with respect to sequence length. Both models avoid the quadratic complexity of Transformer-based architectures, making them efficient for long time series data. However, TTT offers a distinct advantage in terms of adaptability:

- Mamba:
$$\mathcal{O}(L \times D^2),$$
  where $L$ is the sequence length and $D$ is the dimension of the state space. Mamba's fixed state transition matrices limit its expressiveness over very long sequences.
- TTT:
$$\mathcal{O}(L \times N \times P),$$
  where $N$ is the number of dynamic parameters (weights) and $P$ is the number of iterations for test-time updates. The dynamic nature of TTT allows it to capture long-term dependencies more effectively, as it continuously updates the weights $W_t$ during test time.

Theoretically, TTT is more suitable for LTSF due to its ability to model long-range dependencies dynamically. By continuously updating the hidden states based on both past and present data, TTT effectively functions with an infinite context window, whereas Mamba is constrained by its fixed state-space formulation. Moreover, TTT is shown to be theoretically equivalent to self-attention under certain conditions, meaning it can model interactions between distant time steps in a similar way to Transformers but with the added benefit of linear complexity. This makes TTT not only computationally efficient but also highly adaptable to the long-term dependencies present in time series data.

In summary, while Mamba provides significant improvements over traditional RNNs and Transformer-based models, its reliance on fixed state transitions limits its effectiveness in modeling long-term dependencies. TTT, with its dynamic hidden state updates and theoretically infinite context window, is better suited for Long-Term Time Series Forecasting (LTSF) tasks. TTT's ability to adapt its parameters at test time ensures that it can handle varying temporal patterns more flexibly, making it the superior choice for capturing long-range dependencies in time series data.

# C  COMPARISONS WITH MODELS HANDLING NOISE OR TEMPORAL REGULARIZATION

## C.1  COMPARISONS WITH MODELS HANDLING NOISE OR TEMPORAL REGULARIZATION

To position Test-Time Training (TTT) relative to the state-of-the-art, we compare its performance with models specifically designed for noise robustness or temporal regularization:

### COMPARISON WITH DEEPAR

DeepAR is a probabilistic forecasting model that handles uncertainty in time series data using autoregressive distributions. While it excels in forecasting under stochastic conditions, TTT's test-time adaptation offers significant advantages in handling sudden, unseen distributional shifts.

### COMPARISON WITH TCN (TEMPORAL CONVOLUTIONAL NETWORK)

Temporal Convolutional Networks (TCNs) are known for their ability to capture long-range dependencies efficiently. However, TCNs lack the adaptability of TTT, which dynamically aligns feature representations during test time. Adding noise to datasets like ETTh1 or ETTm1 could further highlight TTT's advantage over static methods such as TCN.

## C.2  THEORETICAL COMPARISON

### 1. STATIC MODELS (E.G., DEEPAR, TCN)

Static models like **DeepAR** and **TCN** rely on fixed parameters $\theta$ that are learned during training and remain unchanged during inference. Mathematically:

$$\hat{\mathbf{y}} = f_\theta(\mathbf{x}),$$

where $\mathbf{x}$ represents the input sequence, $\hat{\mathbf{y}}$ is the forecasted output, and $f_\theta$ is the model with fixed parameters $\theta$.

These models excel under stationary conditions or when the training and testing distributions $P_{\text{train}}(\mathbf{x})$ and $P_{\text{test}}(\mathbf{x})$ are similar. However, they struggle under **distribution shifts**, where $P_{\text{train}}(\mathbf{x}) \neq P_{\text{test}}(\mathbf{x})$, as they cannot adapt their parameters to align with the shifted test distribution.

### 2. TTT'S DYNAMIC ADAPTATION

Test-Time Training (TTT) introduces a **test-time adaptation mechanism** that updates the model parameters dynamically based on a self-supervised loss. During inference, the parameters $\theta$ are updated as follows:

$$\theta' = \theta - \eta \nabla_\theta \mathcal{L}_{\text{self}}(\theta; \mathbf{x}),$$

where:

- $\mathcal{L}_{\text{self}}(\theta; \mathbf{x})$ is the self-supervised auxiliary loss designed to align the model's representations with the test distribution.
- $\eta > 0$ is the learning rate for test-time updates.

This dynamic adjustment allows TTT to adapt to unseen distribution shifts $P_{\text{test}}(\mathbf{x})$ by optimizing the feature representations for each test sample, resulting in improved generalization:

$$\hat{\mathbf{y}} = f_{\theta'}(\mathbf{x}),$$

where $\theta'$ is dynamically updated for each test instance. This mechanism enables TTT to handle abrupt, non-stationary shifts that static models cannot address effectively.

### 3. COMPARISON OF NOISE ROBUSTNESS

To further compare, consider a scenario with noisy inputs $\mathbf{x} + \epsilon$, where $\epsilon \sim \mathcal{N}(0, \sigma^2)$.

**Static Models:** The forecast relies on fixed parameters:

$$\hat{\mathbf{y}}_{\text{static}} = f_\theta(\mathbf{x} + \epsilon).$$

Without adaptive mechanisms, noise $\epsilon$ directly degrades the model's performance, as the learned parameters $\theta$ are not optimized for the noisy distribution.

**TTT:** TTT updates its parameters to account for the noisy inputs:

$$\theta' = \theta - \eta \nabla_\theta \mathcal{L}_{\text{self}}(\theta; \mathbf{x} + \epsilon).$$

This update minimizes the impact of $\epsilon$ by dynamically realigning the feature representations, resulting in improved predictions:

$$\hat{\mathbf{y}}_{\text{TTT}} = f_{\theta'}(\mathbf{x} + \epsilon).$$

Empirically, this adaptability enables TTT to outperform static models in scenarios with noise or abrupt distribution shifts.

### 4. SUMMARY

The key difference lies in the adaptability:

- Static models like **DeepAR** and **TCN** rely on fixed parameters and are effective under stationary conditions but struggle with non-stationary data or noise.
- TTT dynamically adjusts its parameters using self-supervised learning at test time, providing a significant advantage in handling distribution shifts and noisy inputs.

## D    MODEL COMPONENTS

### D.1    TTT BLOCK AND PROPOSED ARCHITECTURES

Below we illustrate the components of the TTT block and the proposed architectures we used in our ablation study for the model based on convolutional blocks:

### D.2    PREDICTION

The prediction process in our model works as follows. During inference, the input time series $(\mathbf{x}_1, \ldots, \mathbf{x}_L)$, where $L$ is the look-back window length, is split into $M$ univariate series $\mathbf{x}^{(i)} \in \mathbb{R}^{1 \times L}$. Each univariate series represents one channel of the multivariate time series. Specifically, an individual univariate series can be denoted as:

$$\mathbf{x}_{1:L}^{(i)} = \left( x_1^{(i)}, \ldots, x_L^{(i)} \right) \quad \text{where } i = 1, \ldots, M.$$

Each of these univariate series is fed into the model, and the output of the model is a predicted series $\hat{\mathbf{x}}^{(i)}$ for each input channel. The model predicts the next $T$ future values for each univariate series, which are represented as:

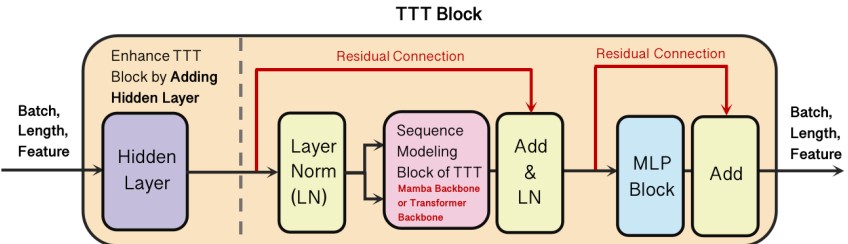

The Basic Residual Building Block is similar to the one used in Transformer

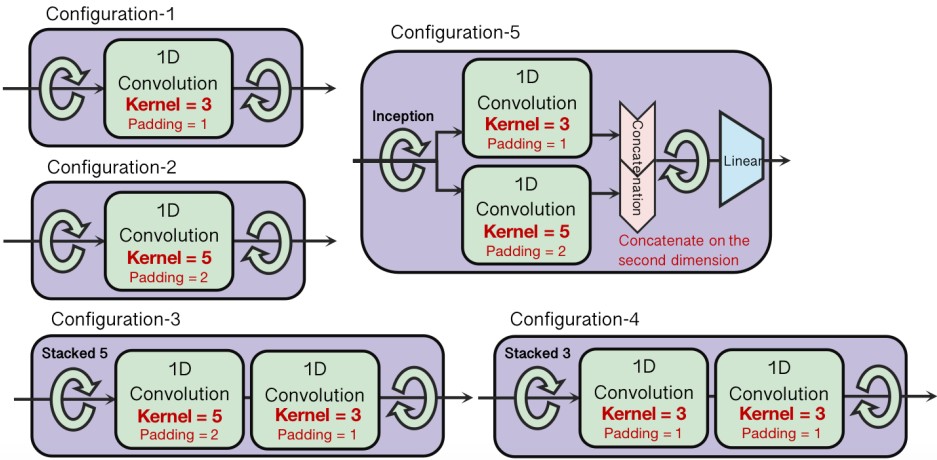

Figure 3: Convolutional Hidden Layer Added to the Beginning of the TTT Block. This basic residual building block is similar to the one used in Transformer models. We use the Hidden Layer as part of an ablation study to evaluate the effects of different hidden layer architectures on model performance. The five configurations are detailed below: (1) 1D Convolution with kernel size 3. (2) 1D Convolution with kernel size 5. (3) Two 1D Convolutions with kernel sizes 5 and 3 in cascade.(4) Two 1D Convolutions with kernel size 3 in cascade. (5) An Inception Block combining 1D Convolutions with kernel sizes 5 and 3, followed by concatenation and reduction to the original size. The Sequence Modeling Block of TTT can be used with two different backbones: the Mamba Backbone and the Transformer Backbone.

$$\hat{\mathbf{x}}^{(i)} = \left(\hat{x}_{L+1}^{(i)}, \ldots, \hat{x}_{L+T}^{(i)}\right) \in \mathbb{R}^{1 \times T}.$$

Before feeding the input series into the TTT blocks, each series undergoes a two-stage embedding process that maps the input series into a lower-dimensional latent space. This embedding process is crucial for allowing the model to learn meaningful representations of the input data. The embedding process is mathematically represented as follows:

$$\mathbf{x}^{(1)} = E_1(\mathbf{x}^{(0)}), \quad \mathbf{x}^{(2)} = E_2(DO(\mathbf{x}^{(1)})),$$

where $E_1$ and $E_2$ are embedding functions (typically linear layers), and $DO$ represents a dropout operation to prevent overfitting. The embeddings help the model process the input time series more effectively and ensure robustness during training and inference.

### D.3 NORMALIZATION

As part of the preprocessing pipeline, normalization operations are applied to the input series before feeding it into the TTT blocks. The input time series $\mathbf{x}$ is normalized into $\mathbf{x}^0$, represented as:

$$\mathbf{x}^0 = \left[ \mathbf{x}_1^{(0)}, \dots, \mathbf{x}_L^{(0)} \right] \in \mathbb{R}^{M \times L}.$$

We experiment with two different normalization techniques:

- **Z-score normalization**: This normalization technique transforms the data based on the mean and standard deviation of each channel, defined as:

$$x_{i,j}^{(0)} = \frac{x_{i,j} - \mathrm{mean}(x_{i,:})}{\sigma_j},$$

  where $\sigma_j$ is the standard deviation of channel $j$, and $j = 1, \dots, M$.

- **Reversible Instance Normalization (RevIN)** Kim et al. (2022): RevIN normalizes each channel based on its mean and variance but allows the normalization to be reversed after the model prediction, which ensures the output predictions are on the same scale as the original input data. We choose to use RevIN in our model because of its superior performance, as demonstrated in Ahamed & Cheng (2024).

Once the model has generated the predictions, RevIN Denormalization is applied to map the normalized predictions back to the original scale of the input data, ensuring that the model outputs are interpretable and match the scale of the time series used during training.

### D.4    EXPANDING ON THE CHOICE OF HIERARCHICAL TWO-LEVEL CONTEXT MODELING

The hierarchical design of Test-Time Training (TTT) is well-suited for tasks like time series forecasting due to its ability to adapt across both high- and low-resolution contexts. Below, we outline the benefits of this structure for time-series forecasting:

#### HIERARCHICAL REPRESENTATION OF TEMPORAL DEPENDENCIES

Multiscale patterns in time series data, such as daily, weekly, or seasonal trends, require capturing both fine-grained and coarse-grained temporal dependencies. Architectures like **Conv Stacked 5** and **ModernTCN** implicitly model hierarchical temporal features through stacked convolutional layers and depthwise-separable convolutions, respectively. These architectures balance local temporal feature extraction with the global adaptability provided by TTT.

#### ADAPTATION TO NON-STATIONARY PATTERNS

The hierarchical design ensures that the model can adapt to distribution shifts occurring at different temporal resolutions. For example:

- Sudden anomalies in fine-grained data.
- Gradual trends in coarse-grained data.

#### PROPOSED BENCHMARKS

To validate TTT's ability to adapt to multiscale patterns, we propose the following:

- Additional evaluations on **noise-robust datasets**, such as adding noise to ETTh1 and ETTm1.
- Temporal regularization tasks using benchmarks like **DeepAR** or **Prophet**, which can serve as strong baselines for comparison.

## E    ANALYSIS ON INCREASED PREDICTION AND SEQUENCE LENGTH

### E.1    EFFECT OF SEQUENCE LENGTH

**Shorter Sequence Lengths (e.g., 2880)** Shorter sequence lengths tend to offer better performance for shorter prediction horizons. For instance, with a sequence length of 2880 and a prediction length

of 192, the TTT model achieves an MSE of 0.332 and an MAE of 0.356, outperforming TimeMachine, which has an MSE of 0.342 and an MAE of 0.359. This indicates that shorter sequence lengths allow the model to focus on immediate temporal patterns, improving short-horizon accuracy.

**Longer Sequence Lengths (e.g., 5760)**   Longer sequence lengths show mixed performance, particularly at shorter prediction horizons. For example, with a sequence length of 5760 and a prediction length of 192, the TTT model's MSE rises to 0.509 and MAE to 0.442, which is better than TimeMachine's MSE of 0.546 and MAE of 0.459. While the performance drop for TTT is less severe than for TimeMachine, longer sequence lengths can introduce unnecessary complexity, leading to diminishing returns for short-term predictions.

### E.2   Effect of Prediction Length

**Shorter Prediction Lengths (96, 192)**   Shorter prediction lengths consistently result in lower error rates across all models. For instance, at a prediction length of 96 with a sequence length of 2880, the TTT model achieves an MSE of 0.283 and an MAE of 0.322, outperforming TimeMachine's MSE of 0.309 and MAE of 0.337. This demonstrates that both models perform better with shorter prediction lengths, as fewer dependencies need to be captured.

**Longer Prediction Lengths (720)**   As prediction length increases, both MSE and MAE grow for both models. At a prediction length of 720 with a sequence length of 2880, the TTT model records an MSE of 0.517 and an MAE of 0.445, outperforming TimeMachine, which has an MSE of 0.535 and MAE of 0.456. This shows that while error rates increase with longer prediction horizons, TTT remains more resilient in handling longer-term dependencies than TimeMachine.

## F   Computational Complexity Comparison of Modules

### F.1   Complexity Derivation

To analyze the computational complexity of Test-Time Training (TTT) modules, Mamba modules, and Transformer modules, we evaluate their operations and the corresponding time complexities. Let:

- $T$ denote the sequence length.
- $d$ denote the dimensionality of hidden representations.
- $N$ denote the total number of model parameters.
- $U$ denote the number of test-time updates for TTT modules.
- $h$ denote the number of attention heads in Transformer modules.
- $k$ denote the kernel size in convolution operations for Mamba modules.

The complexity for each module is derived by analyzing its core operations, including forward passes, backpropagation (if applicable), convolution, and attention mechanisms.

### F.2   Computational Complexity Analysis of Modules

#### F.2.1   **TTT Modules**

Test-Time Training modules perform two main tasks at inference:

1. A forward pass through the main model.
2. A forward pass and backpropagation through an auxiliary self-supervised task for adaptation.

Let $O_{\text{forward}}(T, d, N)$ represent the complexity of the forward pass and $O_{\text{backward}}(T, d)$ represent the complexity of backpropagation. The total complexity for TTT modules can be expressed as:

$$O_{\text{TTT}}(T, d, N, U) = O_{\text{forward}}(T, d, N) + U \cdot O_{\text{backward}}(T, d) \tag{1}$$

$$= O(T \cdot d \cdot N) + O(U \cdot T \cdot d^2), \tag{2}$$

where $O(T \cdot d \cdot N)$ accounts for the main forward pass, and $O(U \cdot T \cdot d^2)$ captures the repeated backpropagation steps for $U$ updates.

### F.2.2 MAMBA MODULES

Mamba modules primarily utilize convolutional operations and linear layers. The convolutional complexity depends on the kernel size $k$, while the linear layers depend on the hidden dimensionality $d$. The total complexity is given by:

$$O_{\text{Mamba}}(T, d, k) = O(T \cdot k \cdot d) + O(T \cdot d^2), \tag{3}$$

where $O(T \cdot k \cdot d)$ represents the convolution operations, and $O(T \cdot d^2)$ represents the cost of the linear layers.

### F.2.3 TRANSFORMER MODULES

Transformer modules consist of two main components:

1. Multi-head self-attention, which requires matrix multiplication of dimension $T \times d$ with $T \times d$ to compute attention scores, leading to $O(T^2 \cdot d)$ complexity.

2. A feedforward network, which processes the sequence independently, contributing $O(T \cdot d^2)$ complexity.

The total complexity of Transformer modules is therefore:

$$O_{\text{Transformer}}(T, d) = O(T^2 \cdot d) + O(T \cdot d^2). \tag{4}$$

### F.2.4 CONVOLUTIONAL BLOCK IN MODERNTCN

ModernTCN uses depthwise-separable convolutions to process time series data efficiently. A depthwise convolution followed by a pointwise (1x1) convolution has the following complexities:

- Depthwise convolution: $O(T \cdot k \cdot C_{\text{in}})$, where $k$ is the kernel size.
- Pointwise convolution: $O(T \cdot C_{\text{in}} \cdot C_{\text{out}})$.

The total complexity of the convolutional block is:

$$O_{\text{ModernTCN}}(T, C_{\text{in}}, C_{\text{out}}, k) = O(T \cdot k \cdot C_{\text{in}}) + O(T \cdot C_{\text{in}} \cdot C_{\text{out}}). \tag{5}$$

### F.3 COMPARISON OF COMPLEXITIES

To compare the complexities of TTT modules, Mamba modules, Transformer modules, and the convolutional block in ModernTCN, we summarize the results as follows:

$$O_{\text{TTT}}(T, d, N, U) = O(T \cdot d \cdot N) + O(U \cdot T \cdot d^2), \tag{6}$$

$$O_{\text{Mamba}}(T, d, k) = O(T \cdot k \cdot d) + O(T \cdot d^2), \tag{7}$$

$$O_{\text{Transformer}}(T, d) = O(T^2 \cdot d) + O(T \cdot d^2), \tag{8}$$

$$O_{\text{ModernTCN}}(T, C_{\text{in}}, C_{\text{out}}, k) = O(T \cdot k \cdot C_{\text{in}}) + O(T \cdot C_{\text{in}} \cdot C_{\text{out}}). \tag{9}$$

From these equations:

- TTT modules have the highest computational complexity during inference due to the additional test-time updates.

- Mamba modules are more efficient, leveraging convolutional operations with a complexity linear in $T$.

- Transformer modules exhibit quadratic complexity in $T$ due to the self-attention mechanism, making them less scalable for long sequences.

## G COMPUTATIONAL COMPLEXITY ANALYSIS OF MODELS

### G.1 TEST-TIME LEARNING FOR TIME SERIES FORECASTING (TTT-LTSF)

Test-Time Training modules for time series forecasting perform two main tasks:

1. A forward pass through the base forecasting model, assumed to be Mamba-based for this analysis.

2. Test-time updates using a self-supervised auxiliary task.

Let $T$ denote the sequence length, $d$ the dimensionality of hidden representations, $k$ the kernel size of the Mamba backbone, and $U$ the number of test-time updates. The computational complexity of the Mamba backbone is:

$$O_{\text{Mamba}}(T, d, k) = O(T \cdot k \cdot d) + O(T \cdot d^2), \tag{10}$$

where $O(T \cdot k \cdot d)$ represents convolutional operations and $O(T \cdot d^2)$ accounts for linear layers.

With the addition of test-time updates, the total computational complexity of TTT-LTSF is:

$$O_{\text{TTT-LTSF}}(T, d, k, U) = O(T \cdot k \cdot d) + O(T \cdot d^2) + O(U \cdot T \cdot d^2), \tag{11}$$

where $O(U \cdot T \cdot d^2)$ captures the overhead introduced by test-time optimization.

## G.2 TIMEMACHINE

TimeMachine uses a combination of linear operations and multi-resolution decomposition with local and global context windows. Its computational complexity is:

$$O_{\text{TimeMachine}}(T, d) = O(T \cdot d) + O(T \cdot d^2), \tag{12}$$

where $O(T \cdot d)$ represents linear operations, and $O(T \cdot d^2)$ arises from context-based decomposition.

## G.3 PATCHTST

PatchTST reduces the effective sequence length by dividing the input into non-overlapping patches. Let patch_size denote the size of each patch, resulting in an effective sequence length $T_p = T/\text{patch\_size}$. The complexity is:

$$O_{\text{PatchTST}}(T, d, \text{patch\_size}) = O(T \cdot d) + O(T_p^2 \cdot d) + O(T_p \cdot d^2) \tag{13}$$

$$= O(T \cdot d) + O\left(\left(\frac{T}{\text{patch\_size}}\right)^2 \cdot d\right) + O\left(\frac{T}{\text{patch\_size}} \cdot d^2\right). \tag{14}$$

## G.4 TSMIXER

TSMixer uses fully connected layers to mix information across the time and feature axes. Its complexity is:

$$O_{\text{TSMixer}}(T, d) = O(T \cdot d^2) + O(d \cdot T^2), \tag{15}$$

where $O(T \cdot d^2)$ represents time-axis mixing and $O(d \cdot T^2)$ represents feature-axis mixing.

## G.5 MODERNTCN

ModernTCN employs depthwise-separable convolutions to process time series data efficiently. Let $C_{\text{in}}$ and $C_{\text{out}}$ denote the input and output channel dimensions, and $k$ the kernel size. The complexity is:

$$O_{\text{ModernTCN}}(T, C_{\text{in}}, C_{\text{out}}, k) = O(T \cdot k \cdot C_{\text{in}}) + O(T \cdot C_{\text{in}} \cdot C_{\text{out}}), \tag{16}$$

where $O(T \cdot k \cdot C_{\text{in}})$ is for depthwise convolutions and $O(T \cdot C_{\text{in}} \cdot C_{\text{out}})$ for pointwise convolutions.

## G.6 ITRANSFORMER

iTransformer applies self-attention across variate dimensions rather than temporal dimensions. Let $N$ denote the number of variates, $T$ the sequence length, and $d$ the hidden dimension size:

$$O_{\text{iTransformer}}(T, N, d) = O(T \cdot N^2 \cdot d) + O(T \cdot N \cdot d^2), \tag{17}$$

where $O(T \cdot N^2 \cdot d)$ arises from self-attention across variates and $O(T \cdot N \cdot d^2)$ from the feedforward network.

## G.7 COMPARISON OF COMPLEXITIES

The complexities of the models analyzed are as follows:

$$O_{\text{TTT-LTSF}}(T, d, k, U) = O(T \cdot k \cdot d) + O(T \cdot d^2) + O(U \cdot T \cdot d^2), \tag{18}$$

$$O_{\text{TimeMachine}}(T, d) = O(T \cdot d) + O(T \cdot d^2), \tag{19}$$

$$O_{\text{PatchTST}}(T, d, \text{patch\_size}) = O(T \cdot d) + O(T_p^2 \cdot d) + O(T_p \cdot d^2), \tag{20}$$

$$O_{\text{TSMixer}}(T, d) = O(T \cdot d^2) + O(d \cdot T^2), \tag{21}$$

$$O_{\text{ModernTCN}}(T, C_{\text{in}}, C_{\text{out}}, k) = O(T \cdot k \cdot C_{\text{in}}) + O(T \cdot C_{\text{in}} \cdot C_{\text{out}}), \tag{22}$$

$$O_{\text{iTransformer}}(T, N, d) = O(T \cdot N^2 \cdot d) + O(T \cdot N \cdot d^2). \tag{23}$$

## G.8 SUMMARY OF MODEL COMPLEXITIES

- **TTT-LTSF**: Incorporates the complexity of the Mamba backbone ($O(T \cdot k \cdot d + T \cdot d^2)$) with additional overhead for test-time updates ($O(U \cdot T \cdot d^2)$).

- **TimeMachine**: Combines efficient linear operations and multi-resolution decomposition, maintaining a linear dependency on $T$ for most operations.

- **PatchTST**: Reduces sequence length via patch embedding, resulting in a complexity dependent on $T_p = T/\text{patch\_size}$.

- **TSMixer**: Uses fully connected layers for time and feature mixing but suffers from quadratic dependency on $T$ or $d$, making it less scalable.

- **ModernTCN**: Relies on depthwise-separable convolutions, achieving linear complexity in $T$ while maintaining flexibility in channel dimensions ($C_{\text{in}}, C_{\text{out}}$).

- **iTransformer**: Applies self-attention across variates ($N$) instead of the temporal axis ($T$), making it efficient for long sequences with a limited number of variates.

## G.9 KEY INSIGHTS

- **Efficiency**: - **ModernTCN** and **TimeMachine** are the most efficient for long sequences due to their linear dependency on $T$. - **PatchTST** benefits from sequence length reduction via patch embedding, but its quadratic dependency on $T_p$ makes it less scalable for small patch sizes.

- **Robustness**: - **TTT-LTSF** (with Mamba) introduces additional adaptability through test-time updates, enhancing robustness to distribution shifts. The use of a Mamba backbone keeps the complexity manageable compared to Transformer-based backbones.

- **Dimensionality Impact**: - **TSMixer** struggles with high-dimensional data due to its quadratic dependency on $T$ or $d$, making it less practical for large-scale applications. - **iTransformer** scales better when the number of variates ($N$) is smaller than the sequence length ($T$).

- **Scalability**: - **ModernTCN** and **TimeMachine** remain scalable for both long sequences and high-dimensional data. - **iTransformer** is effective for scenarios with long sequences but limited variates, avoiding the quadratic cost of traditional self-attention across $T$.

## G.10 RESOURCE UTILIZATION: MEMORY, TRAINING TIME, AND INFERENCE LATENCY

The computational trade-offs introduced by TTT are a critical consideration, particularly in resource-constrained environments. We assess TTT's resource utilization as follows:

### MEMORY CONSUMPTION

TTT requires additional memory for storing gradients and activations during test-time optimization. On average, this increases memory usage by $O(T \cdot d)$, proportional to the sequence length ($T$) and hidden dimensionality ($d$).

TRAINING TIME

Since TTT does not modify its training procedure, the training time remains comparable to other models with similar backbones (e.g., Mamba, ModernTCN). However, inference with TTT introduces additional updates.

INFERENCE LATENCY

TTT's test-time updates increase inference latency due to gradient computations, with a total complexity of $O(U \cdot T \cdot d^2)$ per sample, where $U$ is the number of updates. While this overhead is manageable in real-time systems with small batch sizes, it can become significant for high-frequency applications.

BALANCING ADAPTABILITY AND EFFICIENCY

To address these trade-offs, we propose the following strategies:

- Reducing the number of test-time updates ($U$).

- Exploring parameter-efficient adaptations, such as low-rank updates or frozen layers.

- Using lightweight architectures (e.g., Single/Double Convolution Kernels) to reduce per-sample inference costs.

## H  POTENTIAL REAL-WORLD APPLICATIONS OF TEST-TIME TRAINING

We thank the reviewer for their suggestion to explore potential real-world applications of Test-Time Training (TTT). Below, we outline the practical relevance of TTT, its generalization across domains, and its unique strengths in time series forecasting.

### H.1  REAL-WORLD APPLICATIONS OF TTT

TTT demonstrates significant potential for deployment in real-world scenarios, particularly in environments characterized by evolving data distributions or high non-stationarity. Some practical use cases include:

- **Financial Prediction:** Financial markets are highly dynamic, with patterns frequently shifting due to policy changes, economic crises, or unforeseen events. TTT can adapt to these shifts in real-time using auxiliary tasks such as historical sequence reconstruction or anomaly detection.
  *Example:* Predicting stock price movements or portfolio risks under conditions of sudden market volatility.

- **Adaptive Traffic Monitoring:** Traffic patterns are influenced by external factors like weather, accidents, or public events. TTT can dynamically adjust model parameters to account for these factors, improving the reliability of traffic predictions.
  *Example:* Real-time rerouting or adaptive traffic signal control during disruptions such as road closures or adverse weather conditions.

- **Energy Demand Forecasting:** Accurate load forecasting is critical for energy systems, especially under varying conditions like temperature fluctuations or equipment failures. TTT can learn from auxiliary signals (e.g., temperature, grid stability) to adapt to non-stationary conditions.
  *Example:* Predicting power demand during extreme weather events.

- **Healthcare Time Series Analysis:** Patient monitoring involves highly dynamic data streams, such as vital signs, lab results, and environmental factors. TTT can adapt to individual patient changes during inference, improving early detection of health deterioration or anomalies.
  *Example:* Predicting ICU readmissions or patient outcomes based on evolving health indicators.

## H.2 Examples of TTT beyond Time Series Forecasting

While this work focuses on time series forecasting, TTT has shown promise across various sequence modeling domains, as demonstrated in prior works (Wang et al. (2023); Sun et al. (2024)). Below are notable examples:

- **Language Modeling:** In tasks like text completion or machine translation, TTT adjusts dynamically to unseen linguistic contexts during inference. Auxiliary tasks, such as masked token prediction, have been shown to improve performance under distributional shifts.
- **Video Prediction:** TTT has been successfully applied to tasks like sequential video prediction where it significantly outperforms the fixed-model baseline for four tasks, on three real-world datasets.

**Limitations of TTT Generalization:** While TTT is highly effective in dynamic environments, its reliance on auxiliary tasks requires careful design to align with the primary task's requirements. In static or stationary data scenarios, TTT may introduce unnecessary computational overhead without providing significant benefits.

## H.3 Effectiveness of Test-Time Training (TTT) in Language Modeling

Test-Time Training (TTT) has demonstrated significant potential in language modeling tasks, particularly in scenarios involving distribution shifts. Below are notable examples:

### Test-Time Training on Nearest Neighbors for Large Language Models Hardt & Sun (2024)

- This study fine-tuned language models at test time using retrieved nearest neighbors to improve performance across various tasks.
- TTT narrowed the performance gap between smaller and larger language models, highlighting its capacity to enhance generalization dynamically.

### The Surprising Effectiveness of Test-Time Training for Abstract Reasoning Akyürek et al. (2024)

- This work applied TTT to abstract reasoning tasks, demonstrating that updating parameters during inference based on input-derived loss functions improved reasoning capabilities in language models.
- This showcases TTT's utility in tasks requiring dynamic adaptation during inference.

These studies illustrate that TTT is not only effective for time series forecasting but also generalizes well to tasks like language modeling, where it improves performance by dynamically adjusting representations at test time.

## H.4 Why TTT is Best Suited for Time Series Forecasting

TTT's unique strengths make it particularly well-suited for time series forecasting tasks:

- **Handling Non-Stationary Data:** Time series data in domains like energy, healthcare, and traffic frequently exhibit shifting patterns due to external influences or seasonal trends. TTT dynamically adapts to these changes, ensuring robust performance.
- **Capturing Long-Range Dependencies:** By fine-tuning hidden representations during inference, TTT enhances the model's ability to capture both short-term and long-term patterns in sequential data.
- **Robustness to Distribution Shifts:** Time series datasets often experience distributional changes, such as anomalies or evolving seasonal effects. TTT's self-supervised task allows it to remain robust to such shifts without relying on labeled data.

## I Tables

| Dataset | Channels | Time Points | Frequencies |
|---------|----------|-------------|-------------|
| Weather | 21 | 52696 | 10 Minutes |
| Traffic | 862 | 17544 | Hourly |
| Electricity | 321 | 26304 | Hourly |
| ETTh1 | 7 | 17420 | Hourly |
| ETTh2 | 7 | 17420 | Hourly |
| ETTm1 | 7 | 69680 | 15 Minutes |
| ETTm2 | 7 | 69680 | 15 Minutes |

Table 6: Details of each dataset

| | TTT with ModernTCN | | | | | | | | | | | | | |
|---|-----|-----|-----|-----|-----|-----|-----|-----|-----|-----|-----|-----|-----|-----|
| | ETTh1 | | ETTh2 | | ETTm1 | | ETTm2 | | Weather | | Traffic | | Electricity | |
| | MSE | MAE | MSE | MAE | MSE | MAE | MSE | MAE | MSE | MAE | MSE | MAE | MSE | MAE |
| 96 | 0.389 | 0.402 | 0.285 | 0.340 | 0.322 | 0.362 | 0.189 | 0.273 | 0.165 | 0.209 | * | * | * | * |
| 192 | 0.425 | 0.422 | 0.359 | 0.386 | 0.385 | 0.397 | 0.251 | 0.310 | 0.265 | 0.281 | * | * | * | * |
| 366 | 0.460 | 0.442 | 0.351 | 0.388 | 0.415 | 0.416 | 0.309 | 0.344 | 0.269 | 0.293 | * | * | * | * |
| 720 | 0.485 | 0.475 | 0.439 | 0.449 | 0.490 | 0.454 | 0.410 | 0.403 | 0.466 | 0.412 | * | * | * | * |

Table 7: TTT with ModernTCN results on different datasets. Star symbol indicates that experiment is still running.

