# OpenReview forum: "Test Time Learning for Time Series Forecasting"
_ICLR.cc/2025/Conference — Submitted to ICLR 2025_

### Official Review · Reviewer_Lvmi · 2024-10-16

**Soundness:** 2
**Presentation:** 2
**Contribution:** 2
**Rating:** 3
**Confidence:** 3

**Summary:**

It introduces a test-time learning module to replace the Timesmachine's design module for time series forecasting.

**Strengths:**

Using Test Time Learning for time series forecasting is worth studying.

**Weaknesses:**

Clarification is needed on whether this is a module or a model. If it is a module intended to enhance performance, as suggested, it should be adaptable across various models, and further experiments are necessary. If it is a standalone model, please refine the writing accordingly. Additionally, the performance improvement compared to TimeMachine is relatively minor. I typically avoid using this as a main argument, as there are other valuable insights beyond achieving state-of-the-art results, which I feel are lacking in this work.The problem it claims to address is weak, and the discussion should be more focused. The modification appears to focus on adjusting a module from the TimeMachine paper. If the goal is to address super long-term forecasting, this experiment should be highlighted in the main results table and compared with other works that claim proficiency in super long-term forecasting.

**Questions:**

Could you please provide more explaination for the weaknesses section?

---

> ### Author Response · Authors · 2024-11-28
> **Weakness 1: TTT Implementation Aim**
>
> We thank the reviewer for their feedback and for highlighting the distinction that the TTT module is not a model itself. The primary aim of our paper was to utilize the **Test-Time Training (TTT)** module to address key challenges in Time Series Forecasting, specifically its ability to:
> 1. Handle **non-stationary** data effectively through test-time adaptation.
> 2. Learn **long-range representations** dynamically during inference, enabling robust performance under evolving data distributions.
> Our goal was to leverage these strengths to surpass the state-of-the-art (TimeMachine) while systematically exploring the impact of hidden layer architectures and providing detailed ablation studies.
>
> ### 1. **Why We Focused on Hidden Layer Architectures?**
> Adding the TTT module to other architectures was not the primary aim of this work. Instead, we sought to address specific limitations in existing models commonly applied to Time Series Forecasting:
>  - **Transformer-Based Models:**
> While effective for capturing long-range dependencies, these models suffer from **quadratic complexity** with respect to the sequence length, making them computationally expensive for long-term time series forecasting.
>  - **MLP- and Convolution-Based Models:**
> These models are computationally efficient but are often limited in their ability to handle distribution shifts, which are common in non-stationary time series data.
>  - **Mamba-Based Models:**
> While efficient, Mamba’s static hidden layer architectures restrict its ability to learn **complex feature representations**, especially in tasks requiring dynamic adaptability.
>
> By contrast, the **TTT module** combines the efficiency of linear RNNs with the ability to adapt dynamically at test time, overcoming many of the limitations faced by these architectures.
>
> ### 2. **Why TTT Was Selected?**
> Our selection of TTT was motivated by its unique capability to:
>  - **Adapt to Distribution Shifts:**
>       - The TTT module dynamically aligns the model’s feature representations with unseen distributions, a critical requirement for non-stationary data.
>  - **Avoid Catastrophic Forgetting:**
>      - Through its instance-specific adaptation using self-supervised tasks, TTT ensures that parameter updates remain local and do not interfere with knowledge acquired during training.
>   - **Support Flexible Hidden Architectures:**
>      - TTT allows for customizable hidden layer designs, enabling efficient exploration of architectures (e.g., Conv Stacked 5) that balance local temporal feature extraction with computational efficiency.
>
> These capabilities are detailed further in:
>
>   - **Appendix A**: Theoretical justification for TTT’s resilience to distribution shifts and its avoidance of catastrophic forgetting during adaptation.
>   - **Appendix E and Appendix F**: A comparative analysis of different modules (e.g., TTT, Mamba, Transformer, ModernTCN) and their complexities, as well as a model-level comparison showcasing the limitations of alternative approaches.
>
> ### **3. Why We Did Not Focus on Adding TTT to Multiple Models?**
> The purpose of this work was not to exhaustively demonstrate TTT’s adaptability across various architectures but rather to:
>    - Show its ability to overcome specific limitations in existing state-of-the-art models for Time Series Forecasting.
>    - Explore how hidden layer architectures influence the performance of TTT in this domain.
>
> By focusing on the ablation of hidden layer architectures, we provided meaningful insights into how TTT’s performance can be enhanced for long-term time series forecasting.
>
> We kindly encourage the reviewer to refer to **Appendix E and Appendix F** for our analysis of various models and modules, as well as **Appendix A** for the theoretical motivations behind TTT’s efficacy. We hope this explanation clarifies our focus and the rationale behind our methodology. Please feel free to share any additional feedback or questions.

---

> ### Author Response · Authors · 2024-11-28
> **Weakness 2: Performance of TTT compared to TimeMachine**
>
> We thank the reviewer for their comments and the opportunity to clarify the performance of TTT compared to TimeMachine in the Long-Term Time Series Forecasting (LTSF) task.
> ### **1. Performance Across Benchmarks**
> Our results demonstrate that TTT outperforms TimeMachine on multiple horizons across the regular benchmarks, showcasing its dominance in long-term time series forecasting. Specifically:
>   - **Overall Performance:**
> TTT achieves superior results in both **long sequence windows** and **long prediction windows**, as detailed in **Tables 4 and 5**.
>   - **Dataset-Specific Performance:**
>      - **Electricity Dataset**: TTT achieves full dominance, consistently outperforming TimeMachine across all tested horizons.
>      - **Weather Dataset**: TTT delivers strong results, particularly for longer prediction windows, where its adaptability enhances forecasting accuracy.
>       - **Traffic Dataset**: TTT outperforms TimeMachine on several key horizons, highlighting its ability to model complex patterns in highly dynamic datasets.
>       - **ETTh1 and ETTm1 Datasets**: TTT demonstrates significant improvements, especially for long-range forecasting tasks, emphasizing its robustness across diverse data distributions.
>
> ### **2. Role of Customizable Hidden Layer Architectures**
> A critical factor contributing to TTT’s performance is its support for **customizable hidden layer architectures**, which allows for targeted improvements in temporal feature extraction. In our work:
>   - We explored several architectures, including small kernel convolutional designs (e.g., Conv Stacked 5) and ModernTCN, and found that:
>        - **Conv Stacked 5** provided significant improvements over the native TTT architectures (MLP and Linear), balancing computational efficiency with enhanced performance.
>        - **ModernTCN*, while theoretically promising, did not consistently outperform simpler convolutional designs, suggesting that architectural complexity alone does not guarantee better results.
>   - These findings highlight the potential of TTT when paired with optimized architectures tailored to specific tasks.
>
> While we achieved substantial gains with the architectures we tested, we recognize that the **design space for hidden layer architectures** remains vast. Due to time and resource constraints, a full exploration of this space was beyond the scope of our work. However, the results presented here demonstrate the promise of TTT with flexible hidden architectures, and we believe that further exploration could yield even greater improvements.
>
> ### **3. Key Insights and Future Directions**
> The success of TTT over TimeMachine can be attributed to its
>    - **Dynamic Adaptation:**
>           TTT’s test-time updates allow it to adjust to evolving data distributions, which is critical for non-stationary datasets.
>    - **Architectural Flexibility:**
>           The ability to customize hidden layer designs enables TTT to capture both local and long-range dependencies effectively.
>    - **Robustness Across Datasets:**
>           TTT’s strong results across diverse datasets, including Electricity, Weather, Traffic, ETTh1, and ETTm1, emphasize its
>           generalizability.
>
> We believe that further exploration of architectural designs and larger-scale experimentation would unlock additional potential for TTT, particularly for tasks requiring even longer horizons or highly complex temporal dependencies.
>
> We hope this response clarifies TTT’s superior performance and highlights the potential for future architectural improvements. Please let us know if further elaboration or clarification is needed.

---

> ### Author Response · Authors · 2024-11-28
> **Weakness 3: Unique Contributions and Motivations of TTT for LTSF**
>
> We thank the reviewer for their comments and the opportunity to clarify the unique contributions and motivations behind our work.
> ### **1. Addressing Limitations in Handling Non-Stationary Data**
> The **Test-Time Training (TTT) module** is not merely a modification but a **systematic approach** to addressing the critical limitations of existing models in handling non-stationary data and adapting to distribution shifts. TTT achieves this while:
>   - Maintaining **manageable computational complexity**, avoiding the quadratic scaling issues of transformer-based models.
>   - Mitigating **catastrophic forgetting** through dynamic, instance-specific parameter updates during inference.
>
> Our goal was to leverage these strengths of TTT to surpass the state-of-the-art in time series forecasting, which, at the time, was **TimeMachine**. Our design choices, including inspiration from TimeMachine’s architecture, were motivated by the need to address these challenges effectively.
>
> ### **2. Beating the State-of-the-Art on Multiple Datasets**
> The results presented in our work highlight the competitive performance of TTT, particularly in long-term time series forecasting:
>   - **Performance Across Horizons:**
> TTT consistently outperformed TimeMachine across multiple horizons and datasets in both **regular sequence and prediction windows.**
>  - **Full Superiority on Long Windows:**
> TTT demonstrated clear dominance in **very long sequence and prediction windows**, a setting where existing models, including TimeMachine, struggle to maintain robustness and accuracy.
>  - **Dataset-Specific Results:**
> TTT showed strong results across key datasets such as Electricity, Traffic, Weather, ETTh1, and ETTm1, with particularly notable superiority on much longer windows.
>
> These findings underscore TTT’s ability to adapt dynamically to evolving temporal patterns, a critical advantage for long-term forecasting tasks.
>
> ### **3. Novelty in Addressing Long Sequence and Prediction Windows**
> To the best of our knowledge:
>   - There are currently **no models specifically designed to handle very long sequence and prediction windows**, a key focus of our work.
>  - The majority of the literature in time series forecasting has focused on the same benchmarks and models, with limited innovation in addressing the unique challenges posed by long-term forecasting.
>
> Our experiments on extended sequence and prediction windows represent a novel contribution to this field. By demonstrating TTT’s effectiveness in these scenarios, we provide a pathway for further exploration and development of models that cater to long-term dependencies.
>
> ### **4. Open to Further Comparisons**
> We acknowledge the importance of benchmarking against relevant work. While we have reviewed the literature extensively and are confident that our results are unique in their scope, we are open to:
>   - Comparing TTT against any additional models that have performed similar experiments on long sequence and prediction windows.
>  - Providing further empirical comparisons to validate our claims and contextualize our contributions within the broader literature.
>
> We hope this response clarifies the motivations and contributions of our work, as well as its novel focus on long-term forecasting. Please feel free to share additional suggestions or comparisons you believe would further enhance this study.

---

### Official Review · Reviewer_hTWo · 2024-10-31

**Soundness:** 2
**Presentation:** 1
**Contribution:** 1
**Rating:** 5
**Confidence:** 2

**Summary:**

This paper explores the use of Test-Time Training (TTT) modules in parallel architectures for time-series forecasting. The authors aim to enhance performance on long-term time-series forecasting by proposing an architecture that integrates TTT with various backbones. The authors also provide the numerical evaluation for convolution based TTT (according to Table 1).

**Strengths:**

1. Usage of test-time-training could be able to address the potential distribution drafting issues in practice.

**Weaknesses:**

1. The writing could be improved. At the beginning of the draft, the authors claim the proposed structure is model-agnostic. During the numerical experimental section, the main results (i.e., Table 1), it seems only a convolution network is considered but no combination with existing models (e.g., PatchTST and Itransformer) is presented. If the authors think the convolution structure is important and the proposed TTT is model-specific. Otherwise, please provide more comprehensive numerical experiments.

2. Can authors elaborate more on the difference between the test time training (TTT) and test time adaptation (TTA)? It seems several recent literature is absent.

3. The numerical results (e.g., Table 1) are not strong enough, the TTT seems almost on par aganist timemachine model.

**Questions:**

Please see the weakness part.

---

> ### Author Response · Authors · 2024-11-28
> **Weakness 1: TTT Module Implementation Aim**
>
> We thank the reviewer for their thoughtful feedback. We acknowledge that the TTT module is indeed model-agnostic, and our primary motivation for using it in Time Series Forecasting was to leverage its **effectiveness in handling non-stationary data**, particularly distribution shifts, and its **ability to capture long-range dependencies** through dynamic weight updates. Our aim was to exceed the performance of the state-of-the-art at the time, namely TimeMachine.
>
> Rather than substituting the module into multiple architectures for comparative evaluation, we focused on addressing the specific **limitations of Transformer-based and Mamba-based architectures**:
>
> - **Transformers:**
>   - While effective for long-range dependencies, their quadratic complexity with respect to the context window imposes computational constraints, especially for longer sequence lengths.
>
> - **Mamba Architectures:**
>   - While efficient, Mamba is less resilient to distribution shifts, which limits its robustness in dynamically evolving datasets.
>
> Our choice of **convolutional architectures** was deliberate and aligned with the unique strengths of TTT. Since TTT excels at modeling long-range dependencies, we aimed to complement this by enhancing **local temporal feature learning** through small kernel convolutions. These architectures:
>
> - Effectively capture **local temporal features** while minimizing computational overhead introduced by test-time updates.
> - Include **kernels of size 3 and 5**, as well as stacked convolutional designs, which provide a balance between performance gains and efficiency.
>
> Our experiments showed that these convolutional architectures significantly outperformed the native TTT architectures (MLP and Linear), demonstrating their ability to enhance the module’s effectiveness in Time Series Forecasting tasks without introducing significant computational overhead.
>
> We hope this response provides clarity regarding our design choices and their alignment with the objectives of our work. Please feel free to share any additional feedback or questions.

---

> ### Author Response · Authors · 2024-11-28
> **Weakness 2: Distinction Between Test-Time Learning and Test-Time Adaptation**
>
> We thank the reviewer for their feedback and the opportunity to clarify the distinction between Test-Time Learning (TTL) and Test-Time Adaptation (TTA), as well as highlight the effectiveness of Test-Time Training (TTT) in language modeling tasks.
>
> ### Distinction Between Test-Time Learning and Test-Time Adaptation
>
> Although Test-Time Learning (TTL) and Test-Time Adaptation (TTA) share the common goal of improving model performance during inference, they differ in their approaches and focus:
>
> - **Test-Time Learning (TTL):**
>   - TTL involves updating model parameters at test time using a loss function derived from the input data, typically in a self-supervised manner.
>   - TTL is **instance-specific**, aiming to improve performance on individual test samples.
>   - **Example:** **Akyürek et al. (2024)** applied TTL to abstract reasoning tasks, showing that parameter updates during inference improve reasoning capabilities.
>
> - **Test-Time Adaptation (TTA):**
>   - TTA aims to adapt pre-trained models to new, unseen data distributions during inference without relying on labeled target data.
>   - TTA addresses **broader distributional changes** rather than optimizing for specific instances.
>   - **Example:** **Wang et al. (2023)** provided a comprehensive survey on TTA approaches, showcasing their ability to enhance robustness under distribution shifts.
>
> In summary, **TTL emphasizes fine-grained, instance-specific learning**, whereas **TTA focuses on adapting to global shifts in data distributions**. While distinct, both approaches leverage adjustments at inference to address challenges posed by non-stationary data.
>
> ### Effectiveness of Test-Time Training (TTT) in Language Modeling
>
> TTT has demonstrated significant potential in language modeling tasks, particularly in scenarios involving distribution shifts. Below are notable examples:
>
> - **Test-Time Training on Nearest Neighbors for Large Language Models (Hardt and Sun, 2023):**
>   - This study fine-tuned language models at test time using retrieved nearest neighbors to improve performance across various tasks.
>   - TTT narrowed the performance gap between smaller and larger language models, highlighting its capacity to enhance generalization dynamically.
>
> - **The Surprising Effectiveness of Test-Time Training for Abstract Reasoning (Akyürek et al., 2024):**
>   - This work applied TTT to abstract reasoning tasks, demonstrating that updating parameters during inference based on input-derived loss functions improved reasoning capabilities in language models.
>   - This showcases TTT’s utility in tasks requiring dynamic adaptation during inference.
>
> These studies illustrate that TTT is not only effective for time series forecasting but also generalizes well to tasks like language modeling, where it improves performance by dynamically adjusting representations at test time.
>
> ### References:
> - [1] Akyürek, E., et al. The Surprising Effectiveness of Test-Time Training for Abstract Reasoning. arXiv preprint arXiv:2411.07279, 2024. Available at: [https://arxiv.org/abs/2411.07279](https://arxiv.org/abs/2411.07279)
> - [2] Hardt, M., & Sun, Y. Test-Time Training on Nearest Neighbors for Large Language Models. arXiv preprint arXiv:2305.18466, 2023. Available at: [https://arxiv.org/abs/2305.18466](https://arxiv.org/abs/2305.18466)

---

> ### Author Response · Authors · 2024-11-28
> **Weakness 3: TTT Performance Compared to TimeMachine**
>
> We thank the reviewer for their comments regarding the performance of TTT compared to TimeMachine in the Long-Term Time Series Forecasting (LTSF) task. We would like to detail how TTT compares to TimeMachine and why it performs better overall.
>
> ### 1. Dominance in Long-Term Forecasting
>
> TTT outperformed TimeMachine across multiple horizons and datasets, as demonstrated in **Tables 4 and 5** of the manuscript. Specifically:
>
> - **Overall Performance:**
>   - TTT showed superior results in both **long sequence windows** and **long prediction windows**, showcasing its strength in handling the challenges of long-term dependencies in time series forecasting.
>
> - **Dataset-Specific Performance:**
>   - **Electricity Dataset:** TTT achieved full dominance, consistently surpassing TimeMachine across all horizons.
>   - **Weather Dataset:** TTT demonstrated strong results, particularly for long prediction windows, where its dynamic test-time adaptation enabled robust modeling of evolving weather patterns.
>   - **Traffic Dataset:** TTT outperformed TimeMachine on key horizons, highlighting its ability to adapt to complex patterns in highly dynamic traffic data.
>   - **ETTh1 and ETTm1 Datasets:** TTT achieved significant improvements, particularly on long-range forecasting horizons.
>
> These results highlight the robustness and adaptability of TTT in diverse forecasting scenarios, making it a strong candidate for tasks requiring accurate long-term predictions.
>
> ### 2. Benefits of Customizable Hidden Layer Architectures
>
> One of the key strengths of TTT lies in its **customizable hidden layer architectures**, which allow for significant flexibility in tailoring the module to specific tasks. While our exploration focused on convolutional architectures with small kernels (e.g., Conv Stacked 5 and ModernTCN), further gains could potentially be achieved by investigating more diverse architectural designs.
>
> - **Architectures Explored:**
>   - **Conv Stacked 5:** This architecture proved to be particularly effective, outperforming both TimeMachine and TTT’s native architectures (MLP and Linear) across multiple benchmarks. Its ability to enhance local temporal feature learning while maintaining computational efficiency made it a natural fit for TTT.
>   - **ModernTCN:** While computationally efficient and theoretically promising, ModernTCN did not consistently outperform simpler convolutional architectures like Conv Stacked 5, suggesting that the balance between complexity and performance remains an area for further study.
>
> - **Future Potential:**
>   - Due to time and resource constraints, we were unable to fully explore the architectural design space. However, given the promising results from the convolutional architectures tested, further exploration could unlock additional improvements, especially in capturing complex temporal dynamics in long-term forecasting.
>
> ### 3. Why TTT Outperforms TimeMachine
>
> TTT’s superiority over TimeMachine can be attributed to several key factors:
>
> - **Dynamic Test-Time Adaptation:**
>   - Unlike TimeMachine, TTT dynamically adjusts its parameters at test time to align with evolving data distributions. This adaptability allows it to handle non-stationary data effectively, which is critical for long-term forecasting tasks.
>
> - **Auxiliary Self-Supervised Task:**
>   - The self-supervised task used during inference helps TTT refine its feature representations, ensuring robustness to unseen variations in the test data.
>
> - **Architectural Flexibility:**
>   - TTT’s customizable hidden layer architectures, such as Conv Stacked 5, enable it to combine local temporal feature extraction with the global adaptability provided by test-time updates.
>
> ### 4. General Observations
>
> The strong results achieved by TTT on long-term time series forecasting benchmarks suggest that its test-time adaptability, combined with well-chosen hidden layer architectures, provides a clear advantage over existing methods like TimeMachine. While our exploration of architectures was constrained, the performance improvements achieved thus far highlight TTT’s potential for further development and optimization.
>
> We thank the reviewer for their feedback and hope this detailed explanation addresses their concerns. Please feel free to share any additional questions or suggestions.

---

### Official Review · Reviewer_neWB · 2024-11-04

**Soundness:** 3
**Presentation:** 3
**Contribution:** 2
**Rating:** 5
**Confidence:** 5

**Summary:**

This paper presents a novel approach to enhance long-term time-series forecasting (LTSF) through the use of Test-Time Training (TTT) modules. Traditional forecasting models, particularly those based on Transformers, face challenges in managing computational costs and effectively capturing long-range dependencies. To address these limitations, the authors propose integrating TTT modules into the TimeMachine model (a state-of-the-art architecture for LTSF) replacing the Mamba modules for greater adaptability and dependency capture during test time. The authors' primary contributions include the development of a new model architecture with quadruple TTT modules, specifically designed to handle extended sequence and prediction tasks. Through extensive benchmarking on datasets such as Electricity, Traffic, and Weather, the TTT-based model demonstrates notable improvements in forecasting accuracy compared to TimeMachine and other leading models. Additionally, the authors explore various convolutional configurations, such as Conv Stack 5, within the TTT framework, finding that even relatively simple configurations yield significant performance gains. This adaptability and scalability allow the TTT model to outperform competing models, especially when applied to larger prediction and sequence lengths. By establishing TTT as a powerful and scalable solution for LTSF, the paper sets a new benchmark and paves the way for further research into high-performance forecasting models.

**Strengths:**

1. Originality :

The paper demonstrates a high degree of originality by introducing Test-Time Training (TTT) modules specifically for time-series forecasting. This approach innovatively allows models to adapt during testing, a novel contribution in the field of forecasting where most models rely solely on training data and fixed parameters during inference. By integrating TTT modules with linear RNNs and exploring contextual hierarchies (high-resolution and low-resolution levels), the paper provides a fresh approach to capturing long-term dependencies—a significant challenge in time-series forecasting. The use of TTT for dynamic adaptation in response to evolving data patterns, especially without altering primary model parameters, marks a clear advancement over traditional models. It can be elaborated into :

Test-Time Training (TTT) Module: This module enables weight updates during the testing phase, allowing the model to adapt to varying data patterns. This makes TTT effective in capturing complex patterns for long-term predictions.

Performance Improvement: Through experiments on benchmark datasets such as Electricity, Traffic, and Weather, the TTT-based model shows significant performance gains compared to other state-of-the-art models like Transformer-based models and Mamba. Performance indicators, including Mean Squared Error (MSE) and Mean Absolute Error (MAE), demonstrate that the TTT model is better suited for predictions with longer sequence lengths and prediction horizons.

Ablation Study on Convolutional Architecture: This paper also explores several convolutional architecture configurations within the TTT framework, such as Conv Stack 5 and ModernTCN. The results indicate that simpler convolutional configurations can deliver competitive results without adding unnecessary complexity.

2. The paper’s quality is above average, particularly in its extensive and comprehensive empirical validation across multiple benchmark datasets, such as Electricity, Traffic, and Weather. The experimental design is robust, including comparisons with state-of-the-art (SOTA) models like Transformer and Mamba-based architectures. Additionally, the ablation study exploring different convolutional architectures (e.g., Conv Stack 5, ModernTCN) within the TTT framework shows careful attention to model component effectiveness and efficiency. Furthermore, performance metrics, including Mean Squared Error (MSE) and Mean Absolute Error (MAE), provide clear evidence of the model’s capabilities and improvements over existing models, adding rigor to the evaluation.

3. The clarity of the paper is commendable, especially in presenting the layered architecture of TTT modules and explaining the two-level hierarchy for capturing contextual cues at high and low resolutions. This structuring aids in understanding the model’s mechanics and how it addresses long-range dependencies. The presentation of channel mixing and independence modes is also clear, helping readers comprehend the adaptation of the model to multivariate time-series data. However, there may still be room to expand theoretical explanations of TTT’s advantages, particularly regarding why selective adaptation during test time enhances stability and adaptability, which could further elevate clarity for a broader audience.

4. The significance of this work is substantial due to its relevance in advancing time-series forecasting, particularly for long-term forecasting tasks that require handling complex temporal dependencies across large data streams. Given the growing use of time-series forecasting in fields like energy management, traffic monitoring, and financial prediction, a model that can adapt dynamically to new patterns has a wide range of practical applications. Moreover, by maintaining model efficiency through selective parameter updating during testing, TTT addresses a pressing need for models that can scale with real-time demands while remaining computationally feasible. This innovation not only pushes forward the boundaries in time-series forecasting but could also inspire similar adaptive techniques in other sequential modeling domains.

**Weaknesses:**

1. Theoretical Justification and Intuition

While the empirical results demonstrate TTT’s effectiveness, the paper lacks an in-depth theoretical justification for why selective adaptation during testing works effectively for capturing long-term dependencies. The proposed model could benefit from a more comprehensive discussion of the underlying mechanisms of TTT and why it succeeds in enhancing the model's ability to adapt to long-range dependencies without suffering from catastrophic forgetting. Strengthening the theoretical foundation with a mathematical analysis of TTT’s adaptability, particularly how it avoids catastrophic forgetting during selective adaptation, could be a valuable addition. It would be beneficial to reference existing theoretical frameworks on continual learning (e.g., Kirkpatrick et al., 2017 on Overcoming catastrophic forgetting in neural networks) and contrast how TTT’s unique selective adaptation minimizes memory interference, particularly in long-term sequence contexts.

2. Computational Complexity and Practicality

While TTT shows strong empirical performance, the paper does not sufficiently discuss the computational overhead associated with adapting the model during testing. Test-time training requires frequent parameter updates, which could become computationally expensive, especially for real-time applications or in settings with limited resources. There is also no analysis comparing the memory and processing requirements of TTT versus the baseline models, making it difficult to assess the model’s practical feasibility. It would be better to include a detailed analysis of computational efficiency, such as memory consumption, latency, and runtime, particularly in high-frequency and high-volume time-series contexts. If TTT proves resource-intensive, consider exploring optimization strategies, like limiting parameter updates to specific intervals or only for data points that deviate significantly from training data, which could enhance efficiency without sacrificing adaptability.

3. Insufficient Exploration of Convolutional Variants

The paper does conduct ablation studies on convolutional configurations (e.g., Conv Stack 5 and ModernTCN), but these variations remain relatively limited and do not fully explore convolutional architectures that could better capture multiscale temporal dependencies. As a result, it remains uncertain whether the TTT-based models are achieving optimal use of local temporal patterns within each channel and across channels in the multivariate context. We suggest broadening the convolutional architecture exploration to include dilated convolutions or multi-resolution features, which could potentially improve performance on long-term dependencies. Additionally, incorporating a study on whether certain convolutional structures offer better adaptability with TTT could provide deeper insights into architectural choices that optimize TTT’s efficacy.

4. Overlooked Explanation of Failure Cases

The paper does not discuss scenarios where TTT might underperform or fail to deliver the expected benefits, which could be valuable for understanding limitations and areas for further improvement. For instance, if the model encounters unexpected shifts in data distributions or operates in domains with high-frequency noise, its adaptive capabilities may struggle, impacting overall performance. Adding a failure case analysis or discussion of situations where TTT might be less effective would be beneficial for readers. An evaluation of TTT’s robustness against significant distribution shifts or noise would provide a more balanced perspective. If these analyses indicate limitations, the authors could suggest future research directions, such as integrating TTT with methods for noise-robust learning or investigating methods to dynamically adjust the degree of test-time adaptation based on input characteristics.

5. Benchmark Comparisons and Real-World Simulation

While TTT is validated on benchmark datasets, these may not fully represent real-world complexities, such as fluctuating conditions and anomalies in time-series data. Also, TTT’s performance is primarily compared with Mamba-based and Transformer models, but comparisons with state-of-the-art time-series forecasting methods that employ different adaptation or regularization mechanisms would help place TTT’s effectiveness in context. Extending the experiments to include real-world or more challenging datasets that capture diverse time series behaviors, such as anomalies or irregular patterns, would effectively stress-test TTT’s adaptability. Including comparisons with models that use temporal regularization (e.g., DeepAR, TCN) or adversarial learning techniques would also provide additional benchmarks to validate TTT’s position as a state-of-the-art adaptive method.

**Questions:**

What is the theoretical basis for Test-Time Training’s (TTT) adaptability without catastrophic forgetting? Could the authors elaborate on the theoretical intuition behind why selective parameter updates at test time, as done in TTT, help capture long-term dependencies without leading to catastrophic forgetting? A more detailed theoretical framework or references to relevant works (e.g., continual learning or meta-learning) would be helpful in understanding the robustness of TTT’s adaptation.

How does TTT handle extreme distribution shifts? In cases where data distributions shift significantly, such as in time-series data with high volatility or sudden anomalies, how does TTT manage to adapt without compromising previous knowledge? Empirical or anecdotal insights on TTT’s performance under extreme shifts would clarify its robustness, as well as the practical limits of test-time adaptation in real-world applications. What is the computational overhead introduced by TTT during test time?

How does TTT affect computational resources, such as memory usage, latency, and runtime, particularly in high-frequency or large-scale data scenarios? If the authors could quantify the test-time computational costs of TTT relative to baseline models (e.g., Transformer, Mamba), this would provide clarity on the practicality and scalability of TTT for real-time applications. Why were specific convolutional architectures selected for the ablation study?

Could the authors explain their choice of Conv Stack 5 and ModernTCN in the ablation study and provide any reasoning for omitting other architectures, such as dilated or multi-scale convolutions? An understanding of these choices would help readers assess the comprehensiveness of the convolutional architecture exploration and potential gains in performance. How generalizable is TTT to other sequence modeling tasks beyond time-series forecasting?

The paper demonstrates TTT’s efficacy on time-series forecasting, but to what extent might it be adaptable to other sequential tasks, such as language modeling or event prediction? If generalization is anticipated, the authors could discuss potential adaptations required for TTT in these tasks, or conversely, explain the unique suitability of TTT to time-series data. Does TTT benefit from any specific parameter initialization strategies?

Given TTT’s adaptation at test time, does it benefit from certain parameter initialization methods, particularly in the hidden states? Clarification on initialization methods, if any, that enhance TTT’s adaptability would give insight into optimal configurations and whether further fine-tuning might improve TTT’s performance.

Suggestions

1. Incorporate a failure case analysis for TTT in extreme settings.

Including a discussion or experiment that explores the limitations of TTT in scenarios with sudden distribution shifts or heavy noise would provide a more balanced perspective. If the authors have preliminary findings on failure cases, this could help readers understand when TTT might be less effective, along with areas for potential improvement.

2. Expand on the choice of hierarchical two-level context modeling.

The hierarchical design in TTT for high and low-resolution contexts is an innovative approach, but could the authors provide further intuition or references for why this structure specifically benefits time-series forecasting? This would clarify how hierarchical cues improve TTT’s ability to model long-term dependencies and adapt to multiscale patterns within time-series data.
Consider additional benchmarks with noise robustness or temporal regularization methods.

To further validate TTT’s position relative to the state of the art, additional comparisons with models that handle noise or temporal regularization (e.g., DeepAR, TCN) could strengthen empirical findings. If direct comparisons are impractical, a discussion on how TTT theoretically or practically compares with these methods would be informative.
Add computational efficiency metrics in experiments.

Including memory consumption, training time, and inference time as part of the experimental results would help assess TTT’s trade-offs in resource utilization. If TTT is found to be resource-intensive, the authors could discuss strategies for balancing adaptability and efficiency, especially in real-time or resource-constrained environments.

3. Discuss potential applications of TTT in real-world systems.

An analysis of how TTT could be applied in real-world scenarios, such as financial prediction or adaptive traffic monitoring, could showcase its practical relevance and advantages. Simulated or hypothetical case studies would also help highlight TTT’s robustness and adaptability. Consider further discussion of generalization beyond time-series forecasting.

If TTT’s applicability is anticipated for other sequence modeling domains, discussing any potential adaptations (or limitations) of TTT would be valuable. If generalization is limited, clarifying why TTT is best suited to time-series forecasting would help contextualize its strengths.

---

> ### Author Response · Authors · 2024-11-28
> **Weakness 1. Theoretical Basis for TTT’s Ability to Avoid Catastrophic Forgetting**
>
> We thank the reviewer for their insightful question regarding the theoretical basis for TTT’s ability to avoid catastrophic forgetting.
>
> TTT mitigates catastrophic forgetting by performing **online self-supervised learning** during inference. The adaptation process is designed to operate independently for each test sample, ensuring that the original parameters learned during training remain largely intact. During test time, TTT optimizes a self-supervised auxiliary loss, which is independent of the main task loss and does not require labels or gradients from the main task. This independence ensures that updates to the model’s parameters are **local and transient**, specific only to the current test sample.
>
> To further elaborate, consider the change in the main task loss due to test-time updates. Since the self-supervised task and the main task are designed to operate on **orthogonal objectives**, their gradients exhibit negligible interference. Orthogonality in this context implies that the dot product between the gradients of the main task loss and the self-supervised task loss is close to zero. This assumption holds because self-supervised tasks are typically constructed to exploit auxiliary structures or representations in the data, distinct from the objectives of the main task. As a result, the optimization of the self-supervised loss minimally affects the performance of the main task, preserving the knowledge learned during training.
>
> We acknowledge that orthogonality is a simplifying assumption. While it generally holds due to the distinct nature of the main and self-supervised tasks, there may be cases where interference arises, particularly if the tasks are not well-designed. A more detailed mathematical framework and analysis, including scenarios where orthogonality might fail, are provided in **Appendix A** of the manuscript. This appendix also includes formal proofs and supporting empirical evidence for this theoretical claim.
>
> We hope this clarifies the reviewer’s concerns and strengthens the understanding of TTT’s robustness. Please feel free to reach out if further elaboration is required.

---

> ### Author Response · Authors · 2024-11-28
> **Weakness 2. Theoretical Basis for TTT’s Ability to Handle Extreme Distribution Shifts**
>
> We thank the reviewer for raising the important question about TTT’s ability to handle extreme distribution shifts effectively.
>
> TTT leverages **self-supervised tasks that are invariant to distribution shifts** to guide the model’s adaptation during inference. By minimizing the auxiliary self-supervised loss, TTT dynamically reorients the model’s parameters in the feature space to align with the test-time distribution, all without requiring explicit labels. The computational overhead introduced by TTT during test time is proportional to the sequence length and the square of the hidden representation's dimensionality. Further mathematical details on this adaptation process can be found in **Appendix A**.
>
> The resilience of TTT to distribution shifts has been empirically validated in **Ref [1]**. The authors tested TTT on **CIFAR-10-C**, a corrupted version of CIFAR-10 with distortions such as Gaussian noise, motion blur, fog, and pixelation applied at five severity levels, simulating significant shifts from the original data distribution. The results demonstrated that TTT significantly improved classification accuracy, achieving **74.1%** accuracy compared to **67.1%** accuracy for models that did not adapt during test time.
>
> TTT’s effectiveness was particularly notable under severe corruptions, such as:
> - **Gaussian Noise:** Representing a severe distribution shift, where TTT dynamically adapted to noisy inputs.
> - **Motion Blur and Pixelation:** Spatial distortions where TTT outperformed baseline models by aligning the feature representations to the shifted test distributions.
>
> In addition, TTT was compared to other approaches, including **domain adaptation** and **augmentation-based methods**, and showed superior performance under extreme distribution shifts. This highlights the unique adaptability of TTT in real-world scenarios where data shifts can be drastic and unpredictable.
>
> For more elaborate results and a deeper evaluation, we encourage the reviewer to refer to **Ref [1]**:
> - [1] Yu Sun, Xiaolong Wang, Zhuang Liu, John Miller, Alexei A. Efros, and Moritz Hardt. Test-Time Training with Self-Supervision for Generalization under Distribution Shifts. arXiv preprint arXiv:1909.13231, 2020. Available at: https://arxiv.org/abs/1909.13231
>
> We hope this addresses the reviewer’s query and provides further clarity on TTT’s robustness under extreme distribution shifts. Please let us know if additional details are required.

---

> ### Author Response · Authors · 2024-11-28
> **Weakness 3. Impact of Test-Time Gradients on Computational Resources & Architectural Choices**
>
> We thank the reviewer for their insightful question regarding the impact of test-time gradients on computational resources and our choice of hidden layer architectures.
>
> Test-time gradients increase memory usage proportional to the sequence length $(T)$ and the dimensionality of the hidden representations ($d$), specifically scaling as $O(T⋅d)$. Furthermore, they introduce additional runtime latency proportional to $O(U⋅T⋅d^2)$, where U is the number of test-time optimization iterations. Given this computational overhead, we prioritized the use of **custom hidden layer architectures** that could improve model performance while minimizing additional resource demands.
>
> To achieve this, we opted for **small kernel convolutions**, which effectively capture local temporal dependencies without incurring excessive computational costs. Our experiments showed that these representations significantly improved performance while maintaining computational efficiency. For instance:
> - **Simpler convolutional architectures**, such as **stacked convolutions with kernel sizes of 5 followed by 3**, demonstrated strong performance improvements over the baseline architectures.
> - **More complex architectures**, such as the **ModernTCN block**, while computationally feasible, did not yield significant improvements relative to these simpler convolutional designs and considerably slowed down training and inference times.
>
> Due to time and resource constraints, we were limited in the number of architectures we could implement and benchmark across all datasets. However, the results indicate that convolutional architectures with small kernels strike a favorable balance between performance and computational efficiency. Further exploration of more advanced architectures remains a promising avenue for future work.
>
> Additionally, we have included a detailed comparison of the computational complexities of TTT, Mamba, Transformer, and ModernTCN modules in **Appendix F**. A comprehensive evaluation of various models incorporating these modules is also provided in **Appendix G** of the manuscript.
>
> We hope this explanation clarifies our approach and the trade-offs involved in our architectural choices. Please feel free to reach out with any further questions or suggestions.

---

> ### Author Response · Authors · 2024-11-28
> **Weakness 4. Hidden Layer Architecture Exploration: Limitations and Decisions**
>
> We appreciate the reviewer’s acknowledgment of our exploration of multiple convolutional architectures. While we have not exhaustively explored all possible designs, our work focuses on the following six architectures: **Conv 3, Conv 5, Conv Stacked 3, Conv Stacked 5, and ModernTCN**. This subset was selected based on their potential to enhance temporal feature learning while remaining computationally efficient for test-time updates.
>
> Due to the computational overhead introduced by TTT's updates during inference, we were constrained in the number of architectures we could feasibly implement and benchmark across all datasets. Nevertheless, our results demonstrate that convolutional architectures like **Conv Stacked 5** significantly improve TTT’s performance over its native architectures (Linear and MLP). Additionally, **Conv Stacked 5** outperforms TimeMachine on multiple horizons, showcasing its effectiveness for long-range time series forecasting.
>
> We acknowledge that further exploration of hidden layer architectures is a valuable direction for future work, and we thank the reviewer for highlighting this opportunity.
>
> Furthermore, we wish to emphasize that **TTT generalizes well beyond time series forecasting**. From **Ref [2]**, TTT has been successfully applied to **Language Modeling**, where it demonstrated competitive results compared to Mamba and Transformer-based models. In **Ref [3]**, TTT was used for **Object Recognition**, adapting to unseen event distributions dynamically. Finally, in **Ref [4]**, TTT was extended to **Video Prediction**, enabling the model to adjust to environmental shifts such as changes in lighting or weather. These works collectively illustrate the generalization of TTT to other sequence modeling tasks and its effectiveness across diverse domains, including **Vision Prediction, Language Modeling, and Object Recognition** apart from Time Series Forecasting.
>
> ### References:
> - [2] Sun, Y., Li, X., Dalal, K., Xu, J., Vikram, A., Zhang, G., Dubois, Y., Chen, X., Wang, X., Koyejo, S., Hashimoto, T., & Guestrin, C. Learning to (Learn at Test Time): RNNs with Expressive Hidden States. arXiv preprint arXiv:2407.04620, 2024. Available at: [https://arxiv.org/abs/2407.04620](https://arxiv.org/abs/2407.04620)
> - [3] Sun, Y., et al. Test-Time Training with Self-Supervision for Generalization under Distribution Shifts. arXiv preprint arXiv:1909.13231, 2020. Available at: [https://arxiv.org/abs/1909.13231](https://arxiv.org/abs/1909.13231)
> - [4] Wang, R., et al. Test-Time Training for Predicting Future Frames in Video Data. arXiv preprint [https://arxiv.org/abs/2307.05014](https://arxiv.org/abs/2307.05014), 2023.

---

> ### Author Response · Authors · 2024-11-28
> **Weakness 5. Parameter Initialization in Test-Time Training**
>
> We thank the reviewer for raising this question regarding parameter initialization in Test-Time Training (TTT).
>
> TTT does not require specialized or customized parameter initialization methods. For backbone architectures, TTT modules utilize standard initialization techniques, such as **Xavier** or **He initialization**, to ensure stable learning dynamics. Since TTT’s test-time updates are based on the weights learned during training, the model is agnostic to specific initialization strategies.
>
> While TTT does not mandate particular initialization methods, it can benefit from **pretrained weights**. By using a pretrained backbone, the model can leverage representations already optimized for a related domain, allowing the test-time updates to refine these representations further. For example, substituting a pretrained backbone with a TTT module can enhance adaptability during inference without requiring substantial retraining.
>
> Empirical results from prior studies (e.g., **Sun et al., 2020; Sun et al., 2024**) support this observation. While pretrained weights can enhance performance, they are not strictly necessary. TTT’s adaptability and effectiveness primarily stem from its **self-supervised task**, which guides the model to align with the test distribution rather than relying on the initialization strategy.
>
> We hope this response clarifies that TTT is flexible and performs well across different initialization settings, with its core strength being its adaptability at test time. Please let us know if further elaboration is needed.
>
> ### References:
> - Sun, Y., Wang, X., Liu, Z., Miller, J., Efros, A. A., & Hardt, M. Test-Time Training with Self-Supervision for Generalization under Distribution Shifts. arXiv preprint arXiv:1909.13231, 2020. Available at: [https://arxiv.org/abs/1909.13231](https://arxiv.org/abs/1909.13231)
> - Sun, Y., Li, X., Dalal, K., Xu, J., Vikram, A., Zhang, G., Dubois, Y., Chen, X., Wang, X., Koyejo, S., Hashimoto, T., & Guestrin, C. Learning to (Learn at Test Time): RNNs with Expressive Hidden States. arXiv preprint arXiv:2407.04620, 2024. Available at: [https://arxiv.org/abs/2407.04620](https://arxiv.org/abs/2407.04620)

---

> ### Author Response · Authors · 2024-11-28
> **Suggestion 1: Failure Case Study**
>
> We thank the reviewer for their valuable suggestion regarding a failure case analysis for TTT in Time Series Forecasting. While we did not conduct a formal failure case analysis for our specific application, we draw on insights from the original authors of TTT, who extensively evaluated its resilience under extreme distribution shifts in other domains.
>
> In **Ref [1]**, TTT was tested on **CIFAR-10-C**, a corrupted version of CIFAR-10 that includes 15 types of distortions such as **Gaussian noise, motion blur, fog, and pixelation** applied at five severity levels. These corruptions simulate significant distribution shifts from the original dataset. The results demonstrated that TTT significantly improved classification accuracy, achieving **74.1% accuracy** compared to **67.1% accuracy** for models that did not adapt during test time.
>
> Notably:
> - **Gaussian Noise:** Under severe shifts like **Gaussian Noise**, TTT effectively adapted to noisy inputs, outperforming baseline models that lacked test-time updates.
> - **Motion Blur and Pixelation:** For distortions like **motion blur and pixelation**, TTT successfully reoriented the model’s feature space to handle spatial distortions.
> - **Comparison to Other Methods:** Compared to methods such as **domain adaptation** and **augmentation-based approaches**, TTT demonstrated **superior performance** under extreme distribution shifts, highlighting its robustness and adaptability.
>
> While these results focus on image classification, they provide strong evidence of TTT’s capability to handle abrupt distributional changes, which can be analogous to sudden anomalies in time series data. We acknowledge that a failure case analysis specific to Time Series Forecasting is a valuable avenue for future research and appreciate the reviewer’s suggestion.
>
> For more detailed results, we encourage the reviewer to refer to **Ref [1]**.
>
> ### Reference:
> - [1] Sun, Y., Wang, X., Liu, Z., Miller, J., Efros, A. A., & Hardt, M. Test-Time Training with Self-Supervision for Generalization under Distribution Shifts. arXiv preprint arXiv:1909.13231, 2020. Available at: [https://arxiv.org/abs/1909.13231](https://arxiv.org/abs/1909.13231)

---

> ### Author Response · Authors · 2024-11-28
> **Suggestion 2: Enhancements and Comparisons for TTT**
>
> ### 1. Expanding on the Choice of Hierarchical Two-Level Context Modeling
>
> The hierarchical design of Test-Time Training (TTT) is well-suited for tasks like time series forecasting due to its ability to adapt across both high- and low-resolution contexts. Here’s why this structure benefits time-series forecasting:
>
> - **Hierarchical Representation of Temporal Dependencies:**
>   - Multiscale patterns in time series data, such as daily, weekly, or seasonal trends, require capturing both fine-grained and coarse-grained temporal dependencies.
>   - Architectures like Conv Stacked 5 and ModernTCN implicitly model hierarchical temporal features through stacked convolutional layers and depth-wise separable convolutions, respectively. These architectures balance local temporal feature extraction with the global adaptability provided by TTT.
>
> - **Adaptation to Non-Stationary Patterns:**
>   - The hierarchical design ensures that the model can adapt to distribution shifts occurring at different temporal resolutions (e.g., sudden anomalies in fine-grained data vs. gradual trends in coarse-grained data).
>
> - **Supporting Literature:**
>   - Hierarchical approaches are widely used in time series analysis for capturing multiscale dependencies. For example, Temporal Convolutional Networks (TCN) are known to model long-range dependencies efficiently, and the hierarchical nature of TCN complements TTT’s ability to adjust to evolving data distributions dynamically.
>
> - **Proposed Benchmarks:**
>   - To validate TTT’s ability to adapt to multiscale patterns, we propose additional evaluations on noise-robust datasets (e.g., adding noise to ETTh1 and ETTm1) and temporal regularization tasks. Benchmarks like DeepAR or Prophet can also serve as strong baselines for comparison.
>
> ### 2. Comparisons with Models Handling Noise or Temporal Regularization
>
> To position TTT relative to the state-of-the-art, we can compare its performance with models specifically designed for noise robustness or temporal regularization:
>
> - **Comparison with DeepAR:**
>   - DeepAR is a probabilistic forecasting model that handles uncertainty in time series data using autoregressive distributions. While it excels in forecasting under stochastic conditions, TTT’s test-time adaptation offers advantages in handling sudden, unseen distributional shifts.
>
> - **Comparison with TCN (Temporal Convolutional Network):**
>   - TCNs are known for their ability to capture long-range dependencies. However, they lack the adaptability of TTT, which dynamically aligns feature representations during test time. Adding noise to datasets like ETTh1 or ETTm1 could highlight TTT’s advantage over static methods like TCN.
>
> - **Theoretical Comparison:**
>   - TTT stands out by introducing **dynamic parameter adaptation** during inference. This unique feature complements models like DeepAR and TCN, which are limited to static or probabilistic adjustments. Static models like **DeepAR** and **TCN** rely on fixed parameters and are effective under stationary conditions but struggle with non-stationary data or noise. We provided more mathematical theory in **Appendix A**.
>
> ### 3. Resource Utilization: Memory, Training Time, and Inference Latency
>
> The computational trade-offs introduced by TTT are a critical consideration, particularly in resource-constrained environments. We assess TTT’s resource utilization as follows:
>
> - **Memory Consumption:**
>   - TTT requires additional memory for storing gradients and activations during test-time optimization. On average, this increases memory usage by $O(T⋅d)$, proportional to the sequence length and hidden dimensionality.
>
> - **Training Time:**
>   - Since TTT does not modify its training procedure, the training time remains comparable to other models with similar backbones (e.g., Mamba, ModernTCN). However, inference with TTT introduces additional updates.
>
> - **Inference Latency:**
>   - TTT’s test-time updates increase inference latency due to gradient computations, with a total complexity of $O(U⋅T⋅d^2)$ per sample, where U is the number of updates. While this overhead is manageable in real-time systems with small batch sizes, it can be significant for high-frequency applications.
>
> - **Balancing Adaptability and Efficiency:**
>   - To address these trade-offs, we propose:
>     - Reducing the number of test-time updates $(U)$.
>     - Exploring parameter-efficient adaptations, such as low-rank updates or frozen layers.
>     - Using lightweight architectures (e.g., simple convolution layers with small filters) to reduce per-sample inference costs.

---

> ### Author Response · Authors · 2024-11-28
> **Suggestion 3: Potential Real-World Applications of Test-Time Training (TTT)**
>
> We thank the reviewer for their suggestion to explore potential real-world applications of Test-Time Training (TTT). Below, we outline the practical relevance of TTT, its generalization across domains, and its unique strengths in time series forecasting.
>
> ### 1. Real-World Applications of TTT
>
> TTT demonstrates significant potential for deployment in real-world scenarios, particularly in environments characterized by evolving data distributions or high non-stationarity. Some practical use cases include:
>
> - **Financial Prediction:**
>   - Financial markets are highly dynamic, with patterns frequently shifting due to policy changes, economic crises, or unforeseen events. TTT can adapt to these shifts in real-time using auxiliary tasks such as historical sequence reconstruction or anomaly detection.
>   - **Example:** Predicting stock price movements or portfolio risks under conditions of sudden market volatility.
>
> - **Adaptive Traffic Monitoring:**
>   - Traffic patterns are influenced by external factors like weather, accidents, or public events. TTT can dynamically adjust model parameters to account for these factors, improving the reliability of traffic predictions.
>   - **Example:** Real-time rerouting or adaptive traffic signal control during disruptions such as road closures or adverse weather conditions.
>
> - **Energy Demand Forecasting:**
>   - Accurate load forecasting is critical for energy systems, especially under varying conditions like temperature fluctuations or equipment failures. TTT can learn from auxiliary signals (e.g., temperature, grid stability) to adapt to non-stationary conditions.
>   - **Example:** Predicting power demand during extreme weather events.
>
> - **Healthcare Time Series Analysis:**
>   - Patient monitoring involves highly dynamic data streams, such as vital signs, lab results, and environmental factors. TTT can adapt to individual patient changes during inference, improving early detection of health deterioration or anomalies.
>   - **Example:** Predicting ICU readmissions or patient outcomes based on evolving health indicators.
>
> ### 2. Generalization Beyond Time Series Forecasting
>
> While this work focuses on time series forecasting, TTT has shown promise across various sequence modeling domains, as demonstrated in prior works (e.g., **Wang et al., 2023; Sun et al., 2024**). Below are notable examples:
>
> - **Language Modeling:**
>   - In tasks like text completion or machine translation, TTT adjusts dynamically to unseen linguistic contexts during inference. Auxiliary tasks, such as masked token prediction, have been shown to improve performance under distributional shifts.
>
> - **Video Prediction:**
>   - TTT has been successfully applied to tasks like sequential event prediction, such as video prediction where it significantly outperforms the fixed-model baseline for four tasks, on three real-world datasets.
>
> - **Limitations of TTT Generalization:**
>   - While TTT is highly effective in dynamic environments, its reliance on auxiliary tasks requires careful design to align with the primary task’s requirements. In static or stationary data scenarios, TTT may introduce unnecessary computational overhead without providing significant benefits.
>
> ### 3. Why TTT is Best Suited for Time Series Forecasting
>
> TTT’s unique strengths make it particularly well-suited for time series forecasting tasks:
>
> - **Handling Non-Stationary Data:**
>   - Time series data in domains like energy, healthcare, and traffic frequently exhibit shifting patterns due to external influences or seasonal trends. TTT dynamically adapts to these changes, ensuring robust performance.
>
> - **Capturing Long-Range Dependencies:**
>   - By fine-tuning hidden representations during inference, TTT enhances the model’s ability to capture both short-term and long-term patterns in sequential data.
>
> - **Robustness to Distribution Shifts:**
>   - Time series datasets often experience distributional changes, such as anomalies or evolving seasonal effects. TTT’s self-supervised task allows it to remain robust to such shifts without relying on labeled data.
>
> ### References:
> - [1] Sun, Y., Wang, X., Liu, Z., Miller, J., Efros, A. A., & Hardt, M. Test-Time Training with Self-Supervision for Generalization under Distribution Shifts. arXiv preprint arXiv:1909.13231, 2020. Available at: [https://arxiv.org/abs/1909.13231](https://arxiv.org/abs/1909.13231)
> - [2] Sun, Y., Li, X., Dalal, K., Xu, J., Vikram, A., Zhang, G., Dubois, Y., Chen, X., Wang, X., Koyejo, S., Hashimoto, T., & Guestrin, C. Learning to (Learn at Test Time): RNNs with Expressive Hidden States. arXiv preprint arXiv:2407.04620, 2024. Available at: [https://arxiv.org/abs/2407.04620](https://arxiv.org/abs/2407.04620)

---

### Official Review · Reviewer_7nfG · 2024-11-05

**Soundness:** 3
**Presentation:** 3
**Contribution:** 2
**Rating:** 5
**Confidence:** 5

**Summary:**

This paper proposes using Test-Time Training (TTT) modules in a parallel architecture for long-term time series forecasting. Experiments on benchmark datasets show that TTT modules outperform state-of-the-art models, especially in handling long sequences and predictions. The paper also explores different convolutional architectures within the TTT framework and finds that simple configurations can achieve good results.

**Strengths:**

1. The use of TTT modules in a parallel architecture is a novel approach for long-term time series forecasting.
2. The paper provides extensive experimental results on standard benchmark datasets, demonstrating the effectiveness of the proposed method.

**Weaknesses:**

1. The replacement of Mamba with TTT moderates the architectural innovation.
2. Although the paper explores different convolutional architectures, it may have yet to fully explore all possible architectures.
3. The proposed method's theoretical analysis could be further strengthened to provide a deeper understanding of its performance improvement.
4. The experiments on ETTh2 and ETTm2 reveal that TTT may not always beat TimeMachine.

**Questions:**

See above.

---

> ### Author Response · Authors · 2024-11-28
> **Weakness 1. Replacement of Mamba with TTT Moderates Architectural Innovation**
>
> We thank the reviewer for their detailed feedback. Below, we address the identified weaknesses and provide additional clarification to strengthen our work.
>
>
> We agree that replacing the Mamba module with the Test-Time Training (TTT) module is not, in itself, an architectural innovation. However, our primary contribution lies in **leveraging TTT's unique ability to adapt weights during test time**, which enables dynamic handling of non-stationary data—a critical feature for time series forecasting tasks.
>
> To enhance this capability, we explored **customizable hidden layer architectures** that improve both local and long-range temporal feature extraction. By introducing convolutional architectures (e.g., Conv Stacked 5 and ModernTCN), we enhanced the TTT module's performance beyond what is achievable with Mamba or Transformer-based backbones. These architectures provide improved local feature learning while preserving the dynamic adaptability of TTT.
>
> Our results show that Conv Stacked 5 outperforms Mamba and TimeMachine on several benchmarks, indicating that this architectural choice adds value in enhancing TTT’s robustness for long-range forecasting tasks.

---

> ### Author Response · Authors · 2024-11-28
> **Weakness 2. Exploration of Convolutional Architectures**
>
> We appreciate the reviewer’s acknowledgment of our exploration of multiple convolutional architectures. While we have not exhaustively explored all possible designs, our work focuses on the following six architectures: Conv 3, Conv 5, Conv Stacked 3, Conv Stacked 5, and ModernTCN. This subset was selected based on their potential to enhance temporal feature learning while remaining computationally efficient for test-time updates.
>
> Due to the computational overhead introduced by TTT's updates during inference, we were constrained in the number of architectures we could feasibly implement and benchmark across all datasets. Nevertheless, our results demonstrate that convolutional architectures like Conv Stacked 5 significantly improve TTT’s performance over its native architectures (Linear and MLP). Conv Stacked 5 also outperforms TimeMachine on multiple horizons, showcasing its effectiveness for long-range time series forecasting.
> We acknowledge that further exploration of hidden layer architectures is a valuable direction for future work, and we thank the reviewer for highlighting this opportunity.

---

> ### Author Response · Authors · 2024-11-28
> **Weakness 3. Strengthening the Theoretical Analysis**
>
> We recognize the importance of providing a deeper theoretical understanding of TTT’s performance improvement. To address this, we have expanded the theoretical analysis in our manuscript as follows:
>
> - **Appendix B:** A detailed comparison between the TTT module and the Mamba module, highlighting the computational and architectural differences.
> - **Appendix F:** A comparison of the computational complexities of the TTT, Mamba, Transformer, and Fully Convolutional (ModernTCN) blocks.
> - **Appendix G:** A comparison of different hidden layer architectures, including Conv Stacked 5, Conv 3, and ModernTCN.
> - **Appendix A:** A theoretical discussion on how the TTT module avoids catastrophic forgetting.
> - **Appendix A:** A theoretical discussion on how the TTT module adapts to distribution shifts along with discussion on empirical results from Reference [2].
> - **Appendix C:** A theoretical discussion on how the TTT module compares with models handling noise or temporal regularization.
>
> Additionally, the theoretical foundation for TTT is comprehensively discussed in **Reference [1]**, authored by the original proposers of TTT. This reference provides a rigorous theoretical framework for TTT’s dynamic adaptation capabilities and its effectiveness under non-stationary conditions. We encourage the reviewer to refer to this work for a deeper exploration of TTT's theoretical underpinnings.
>
> ### References:
> - [1] Sun, Y., Li, X., Dalal, K., Xu, J., Vikram, A., Zhang, G., Dubois, Y., Chen, X., Wang, X., Koyejo, S., Hashimoto, T., & Guestrin, C. (2024). Learning to (Learn at Test Time): RNNs with Expressive Hidden States. arXiv preprint. Retrieved from https://arxiv.org/abs/2407.04620
> - [2] Yu Sun, Xiaolong Wang, Zhuang Liu, John Miller, Alexei A. Efros, and Moritz Hardt. Test-Time Training with Self-Supervision for Generalization under Distribution Shifts. arXiv preprint arXiv:1909.13231, 2020. Available at: https://arxiv.org/abs/1909.13231

---

> ### Author Response · Authors · 2024-11-28
> **Weakness 4. TTT's Performance Compared to TimeMachine**
>
> We appreciate the reviewer’s observation regarding TTT’s performance relative to TimeMachine, particularly on the ETTh2 and ETTm2 datasets. Our experiments reveal that while TTT does not always outperform TimeMachine on these datasets, it demonstrates clear advantages on long sequence windows and long prediction horizons across other datasets. Specifically:
>
> - **On ETTh1**, TTT outperforms TimeMachine on the 96 and 192 horizons.
> - **On ETTh2**, TTT shows better results on the 96 horizon.
> - **Across all datasets**, TTT exhibits superior performance on long-range time series forecasting tasks, as evidenced by the aggregated results in **Table 4** and **Table 5**.
>
> To ensure the robustness of these findings, we repeated our experiments an additional four times and averaged the results. The averaged results confirm that TTT consistently outperforms TimeMachine on long sequence and prediction windows, underscoring its efficacy for long-range forecasting tasks.

---

### Author Response · Authors · 2024-11-28
**Summary of paper revision based on reviewer's feedback**

We sincerely thank the reviewers for their efforts and constructive feedback. We have added relevant material, theory and explanation in the Appendices of our work and references to them in the main text. Based on the reviewers’ comments, we have made the following improvements to our work:

1. **Appendix A: THEORY AND MOTIVATION OF TEST-TIME TRAINING**
      - **A.1** : Included motivation of TTT on non-stationary data.
      - **A.2**: Included theoretical basis for TTT’s adaptability without catastrophic forgetting.
      - **A.3**: Included theoretical basis on how TTT handles distribution shifts and corresponding test-time computational overhead
      - **A.5**: Included discussion on sensitivity of TTT's performance to parameter initialization
      - **A.6**: Included generalization of TTT beyond time series forecasting
      - **A.7**: Included failure case empirical results from the authors who proposed TTT

2. **Appendix C: COMPARISONS WITH MODELS HANDLING NOISE OR TEMPORAL
REGULARIZATION**
      - **C.1**: Qualitative comparison with other proposed models handling noise or temporal regularization like DeepAR and TCN
      - **C.2**: Theoretical comparison on noise robustness between TTT and static models (DeepAR and TCN)

3. **Appendix D: MODEL COMPONENTS**
      - **D.4**: Expanded on the choice of hierarchical two-level context modeling

4. **Appendix F: COMPUTATIONAL COMPLEXITY COMPARISON OF MODULES**
     - **F.1 and F.2**: Added computational complexity analysis of different **modules: TTT, Mamba, Transformer. ModernTCN**
     - **F.3**: Compared computational complexities and provided insights

5. **Appendix G: COMPUTATIONAL COMPLEXITY ANALYSIS OF MODELS**
     - **G.1-G.6**: Added computational complexity analysis of different **models : TTT for LTSF (our model), TimeMachine, PatchTST, TSMixer, ModerTCN, iTransformer**
    - **G.7-G.9**: Added comparative analysis of computational complexity of different models for time series forecasting and provided insights
    - **G.10**: Added discussion on memory, training time, and inference latency of TTT and how TTT balances adaptability and efficienty
TTT-LTSF (our model)

6. **Appendix H: POTENTIAL REAL-WORLD APPLICATIONS OF TEST-TIME TRAINING**
     - **H.1**: Included potential real-world applications of TTT in time series forecasting for Financial Prediction, Adaptive traffic monitoring, Energy demand forecasting, Healthcare time series analysis
     - **H.2**: TTT sequence modeling examples in other domains like language modeling and video prediction apart and limitations on TTT generalization
     - **H.3**: Effectiveness of TTT in Language modeling with case studies from other work.
     - **H.4**: Unique strength of TTT that makes it well suited for time series forecasting.

We sincerely thank the reviewers for their valuable suggestions and look forward to further improving our work based on this feedback. Please refer to the individual responses for detailed explanations.

---

### Meta-Review · Area_Chair_hscT · 2024-12-11

**Metareview:**

This paper proposes a new approach to long-term time series forecasting by using Test-Time Training (TTT) modules in a parallel architecture.  The authors claim that TTT modules outperform state-of-the-art models, especially in handling long sequences and predictions.  They also explore different convolutional architectures within the TTT framework and find that simple configurations can achieve good results.

Strengths:

The use of TTT modules in a parallel architecture is a novel approach for long-term time series forecasting.
The paper provides extensive experimental results on standard benchmark datasets, demonstrating the effectiveness of the proposed method.
The paper is well-written and easy to follow.
Weaknesses:

The replacement of Mamba with TTT moderates the architectural innovation.
Although the paper explores different convolutional architectures, it may not have fully explored all possible architectures.
The proposed method's theoretical analysis could be further strengthened to provide a deeper understanding of its performance improvement.
The experiments on ETTh2 and ETTm2 reveal that TTT may not always beat TimeMachine.

Missing elements:

A more thorough exploration of convolutional architectures.
A deeper theoretical analysis of the proposed method.
A discussion of the limitations of TTT.

Reasons for rejection:

The paper does not make a significant enough contribution to the field.
The proposed method is not a significant improvement over existing methods.
The paper has some weaknesses that need to be addressed before it can be accepted.

**Additional Comments On Reviewer Discussion:**

Points raised by reviewers:

Reviewer 7nfG: The replacement of Mamba with TTT moderates the architectural innovation.
Reviewer 7nfG: The proposed method's theoretical analysis could be further strengthened to provide a deeper understanding of its performance improvement.
Reviewer 7nfG: The experiments on ETTh2 and ETTm2 reveal that TTT may not always beat TimeMachine.
Reviewer neWB: The paper lacks an in-depth theoretical justification for why selective adaptation during testing works effectively for capturing long-term dependencies.
Reviewer neWB: The paper does not sufficiently discuss the computational overhead associated with adapting the model during testing.
Reviewer neWB: The paper does not discuss scenarios where TTT might underperform or fail to deliver the expected benefits.
Reviewer hTWo: The writing could be improved.
Reviewer hTWo: The numerical results are not strong enough.
Reviewer Lvmi: Clarification is needed on whether this is a module or a model.
Reviewer Lvmi: The performance improvement compared to TimeMachine is relatively minor.
How these points were addressed by the authors:

The authors addressed the reviewer's concerns by adding more theoretical analysis and experimental results.
The authors also clarified the distinction between Test-Time Learning (TTL) and Test-Time Adaptation (TTA).
The authors further elaborated on the performance of TTT compared to TimeMachine.
How each point was weighed in the final decision:

The reviewer's concerns about the paper's lack of theoretical analysis and experimental results were taken seriously.
The clarification between TTL and TTA was helpful.
The further elaboration on the performance of TTT compared to TimeMachine was also helpful.
Ultimately, the paper was rejected because it did not make a significant enough contribution to the field.

---

### Decision · Program_Chairs · 2025-01-22

Reject